# Footprint evidence of early hominin locomotor diversity at Laetoli, Tanzania

Ellison J. McNutt[1,2 ✉], Kevin G. Hatala[3], Catherine Miller[4,5], James Adams[6,7], Jesse Casana[4,5], Andrew S. Deane[8], Nathaniel J. Dominy[4,5], Kallisti Fabian[9], Luke D. Fannin[4,5], Stephen Gaughan[7], Simone V. Gill[10], Josephat Gurtu[9], Ellie Gustafson[11], Austin C. Hill[5,12], Camille Johnson[5], Said Kallindo[9], Benjamin Kilham[13], Phoebe Kilham[13], Elizabeth Kim[2], Cynthia Liutkus-Pierce[14], Blaine Maley[15], Anjali Prabhat[5], John Reader[16], Shirley Rubin[17], Nathan E. Thompson[18], Rebecca Thornburg[11], Erin Marie Williams-Hatala[3], Brian Zimmer[14], Charles M. Musiba[11,19,20] & Jeremy M. DeSilva[4,5,19]

Bipedal trackways discovered in 1978 at Laetoli site G, Tanzania and dated to 3.66 million years ago are widely accepted as the oldest unequivocal evidence of obligate bipedalism in the human lineage[1–3]. Another trackway discovered two years earlier at nearby site A was partially excavated and attributed to a hominin, but curious affinities with bears (ursids) marginalized its importance to the paleoanthropological community, and the location of these footprints fell into obscurity[3–5]. In 2019, we located, excavated and cleaned the site A trackway, producing a digital archive using 3D photogrammetry and laser scanning. Here we compare the footprints at this site with those of American black bears, chimpanzees and humans, and we show that they resemble those of hominins more than ursids. In fact, the narrow step width corroborates the original interpretation of a small, cross-stepping bipedal hominin. However, the inferred foot proportions, gait parameters and 3D morphologies of footprints at site A are readily distinguished from those at site G, indicating that a minimum of two hominin taxa with different feet and gaits coexisted at Laetoli.

In 1976, Peter Jones and Philip Leakey discovered five consecutive bipedal footprints at Laetoli site A within locality 7, a 490 m$^2$ area dated to 3.66 million years ago (Ma) and featuring 18,400 animal tracks[1–3] (Fig. 1). Mary Leakey tentatively suggested that the trackway was made by a hominin[1]. "The footprints," she wrote, "indicate a rolling and probably slow-moving gait, with the hips swivelling at each step, as opposed to the free-striding gait of modern man [humans]." Leakey and Hay[2] classified the footprints as Hominidae, but with a caveat that "the gait was somewhat shambling, with one foot crossing in front of the other."

Unequivocal hominin footprints were discovered at site G two years later, casting doubt on the hominin status of those at site A[3–5]. Researchers described the footprints at site A as "most unusual,"[6] "curiously shaped,"[7] and "enigmatic,"[8] and yet consensus was uniform: they were produced by a plantigrade mammal moving bipedally.

Tuttle[4] advanced three hypotheses to account for the morphology of the footprints and cross-stepping gait (that is, when a foot from each side crosses the midline before touchdown): (1) substrate distortion; (2) they were left by a juvenile bear (ursid); or (3) they are evidence of another hominin species. To test the second possibility, Tuttle[4,9–12] collected data from circus bears trained to walk bipedally and found that their short steps and relatively wide feet were a close match to the site A footprints, although bipedal bears take wider steps. Furthermore, the fifth digit is typically the largest in ursids, solving the 'cross-stepping problem', although Tuttle[4,12] noted that humans do occasionally cross-step. He concluded that "until detailed, naturalistic biometric and kinesiological studies are performed on bipedal bears and barefoot humans, we will defer choosing among the hominid and ursid hypotheses on Laetoli individual A"[4].

Complicating matters further, the internal morphology of the site A footprints was never fully cleaned of matrix infill[1,2,4,5,12]. White and Suwa[8] argued that "reliable identification of these enigmatic prints at Laetoli site A will be impossible until they are more fully cleaned and followed laterally". Accordingly, we were motivated to relocate and re-excavate site A and conduct a detailed comparative analysis of the prints as well as the locomotion of bears (*Ursus americanus*), chimpanzees (*Pan troglodytes*) and humans to test whether the footprints at site A were left by a hominin or an ursid.

[1]Department of Biomedical Sciences, Ohio University Heritage College of Medicine, Athens, OH, USA. [2]Department of Integrative Anatomical Sciences, Keck School of Medicine, University of Southern California, Los Angeles, CA, USA. [3]Department of Biology, Chatham University, Pittsburgh, PA, USA. [4]Ecology, Evolution, Environment and Society Graduate Program, Dartmouth College, Hanover, NH, USA. [5]Department of Anthropology, Dartmouth College, Hanover, NH, USA. [6]Dartmouth Library, Dartmout College, Hanover, NH, USA. [7]Information, Technology, and Consulting, Dartmouth College, Hanover, NH, USA. [8]Department of Anatomy, Cell Biology and Physiology, Indiana University School of Medicine, Indianapolis, IN, USA. [9]Department of Cultural Heritage, Ngorongoro Conservation Area Authority, Arusha, Tanzania. [10]Department of Occupational Therapy, Boston University, Boston, MA, USA. [11]Department of Anthropology, University of Colorado, Denver, CO, USA. [12]Department of Anthropology, University of Pennsylvania, Philadelphia, PA, USA. [13]Kilham Bear Center, Lyme, NH, USA. [14]Department of Geological and Environmental Sciences, Appalachian State University, Boone, NC, USA. [15]Department of Anatomy, Idaho College of Osteopathic Medicine, Meridian, ID, USA. [16]Department of Anthropology, University College London, London, UK. [17]Department of Anthropology, Napa Valley College, Napa, CA, USA. [18]Department of Anatomy, NYIT College of Osteopathic Medicine, Old Westbury, NY, USA. [19]Evolutionary Studies Institute, University of the Witwatersrand, Johannesburg, South Africa. [20]Instituto Superior Politécnico de Tecnologia e Ciências, Luanda Angola, Angola. ✉e-mail: ejmcnutt@ohio.edu

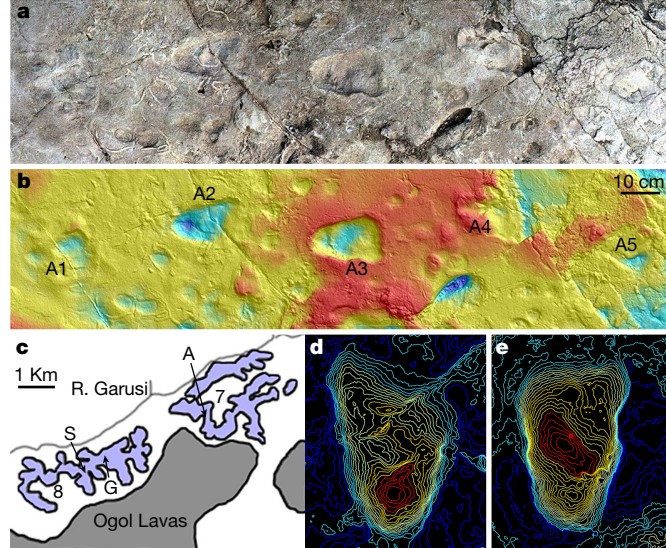

**Fig. 1 | Laetoli location and site rediscovery. a**, A model of site A generated using photogrammetry showing the five hominin footprints. **b**, Corresponding contour map of the site generated from a 3D surface scan with scale bar. **c**, Map of Laetoli localities 7 and 8, indicating the positions of bipedal trackways A, G and S (redrawn from ref. [49]). **d**, **e**, Topographical maps of the two best preserved A footprints, A2 (**d**) and A3 (**e**).

## Rediscovery of site A

Using detailed maps from Leakey and Harris[3], we identified the proboscidean trail adjacent to the bipedal footprints. We cleared the surrounding overburden until one of us (K.F.) found the A3 footprint. The area was then brushed clean to expose A1–A5, which have experienced no discernible erosion since their initial discovery (Fig. 1, Extended Data Fig. 1). Because the footprint tuff is eroded to the north, we excavated south (87 cm) and east (54 cm) from the heel of A1, but no additional footprints were found (Supplementary Information).

After brushing sediment from A3, we used a wooden tongue depressor to remove tuff infill left intact during the 1976–1978 field seasons. The hallucial impression is clearly defined and is about 30 mm wide. Crucially, we exposed the impression of the second digit (Extended Data Fig. 2). We removed infill from A2 but could not do so completely without risking damage. Nevertheless, the heel and hallucial impression are clear. Detailed information from the other footprints (A1, A4 and A5) is limited to estimates of length, width and step length.

Although preservation quality varies within and between A1–A5, there is no evidence that biologically informative metrics were affected by substrate distortion. Adjacent and comingled tracks of other animals (ranging in size from guinea fowl to elephants) show no evidence of distortion to their perimeters or internal morphologies. Given that track surfaces are likely to represent time scales of hours to days[13–15], it is parsimonious to infer similar substrate conditions and taphonomic processes during print formation and subsequent epochs.

## Evaluating ursid and hominin hypotheses

We recorded 50.9 h of video of wild American black bear behaviour. Unsupported bipedal posture and locomotion occurred only 0.09% of the total observation time, of which 59% was postural and 41% was locomotor (Extended Data Fig. 3). In only one instance did a bear take four unassisted bipedal steps. Thus, assuming that our findings are generalizable to other ursids, the probability of observing four consecutive bipedal steps is 0.003%. The low frequency of this behaviour, and the absence of quadrupedal–bipedal transitional footsteps, makes it unlikely, but not impossible, that ursid bipedalism was preserved at site A. Further, Laetoli

is devoid of ursid fossils despite the recovery of more than 25,000 fossils attributed to 85 mammalian species[16,17]. If present at all, ursids were rare on the landscape. Although footprint assemblages can include a surprising number of tracks from taxa whose skeletal fossils are rare (for example, relatively high frequencies of bird tracks at Laetoli[3] and at 1.5-Ma sites near Ileret, Kenya[14,18]), there is no clear taphonomic explanation for why ursid tracks would be present but their fossils absent.

In addition, we measured 46 footprints from four bipedally walking wild juvenile black bears specifically chosen because their foot lengths (mean = 145.7 mm) were within 10% of the length of the site A footprints (mean = 161.7 mm). Additionally, we measured the footprints of chimpanzees produced during quadrupedalism (n = 54 from 46 adults; Ngamba Island Chimpanzee Sanctuary, Uganda) and during bipedalism (n = 44 from two subadults; Stony Brook University, USA). We compared these data with human barefoot footprints produced under three conditions: (1) habitually shod (n = 654) walking on a plantar pressure mat[19]; (2) habitually unshod or minimally shod (n = 41) walking in deformable mud[20,21]; and (3) Late Pleistocene tracks (n = 113) from Engare Sero, Tanzania, formed in reworked volcanic ash[22,23] (summary in Extended Data Table 1).

We concur with others[4,12] that the ratios of footprint dimensions (for example, heel and forefoot width) to step length observed at site A fall within the ursid range (Extended Data Fig. 4a, b). Yet, for these same measures, site A is also chimpanzee-like and moderately similar to definitive hominin footprints from sites G and S. It follows that the site A individual was taking short steps—as occurs when humans walk slowly or over a slippery substrate[24]—not that the gait was ursid-like.

With additional infill removed from A2 and A3, the perimeter dimensions are decidedly hominin-like with wide heel impressions relative to forefoot width (Extended Data Fig. 4c). By contrast, chimpanzees and bears have relatively narrow heels. Furthermore, with the tracks fully excavated and cleaned, we found no evidence for claw impressions, although they are sometimes absent from ursid footprints[12]. Here, impressions were absent from 31% of ursid footprints. To test whether A3 was produced by a hominin left foot or an ursid right foot, we compared the width of the hallux to the second digit in human (n = 30) and chimpanzee (n = 50) footprints, and the fifth digit to the fourth in bear (n = 5) tracks. The A3 toe impressions match the distinctive proportions of humans and chimpanzees rather than those of bears (Extended Data Fig. 5).

By establishing that A3 is a left hominin foot, we can now confirm that cross-stepping occurred. Cross-stepping was never observed in our comparative sample, but humans do it occasionally[25,26] as a compensatory strategy for re-establishing balance after a perturbation[27]. In fact, we suggest that cross-stepping supports the hypothesis that the site A footprints were left by a hominin. Cross-stepping is improbable, and perhaps impossible, when bears or chimpanzees walk bipedally. They produce large mediolateral excursions of their centre of mass[28] and walk with highly abducted hips[29], resulting in a high ratio between stride width and step length. Conversely, human cross-stepping is enabled by their reduced mediolateral centre of mass and body motions, adducted hips and bicondylar angle (that is, valgus knees), resulting in a low corresponding ratio, as expressed in every trackway at Laetoli.

The relative step widths of footprints from sites G and S fall squarely within the modern human distribution (Extended Data Figs. 4d, 5). The site A footprints lie outside the distributions of humans, chimpanzees and bears but are most like humans. This result indicates that the maker of the site A footprints had either valgus knees or adducted hips, or both. The presence of either bipedal trait argues for a hominin maker of the site A footprints.

## Which hominin?

It is generally accepted that *Australopithecus afarensis* produced the footprints at sites G and S[8] (but see refs. [12,30–32]). It is thus tempting to assign the site A tracks to *A. afarensis*; however, this premise requires

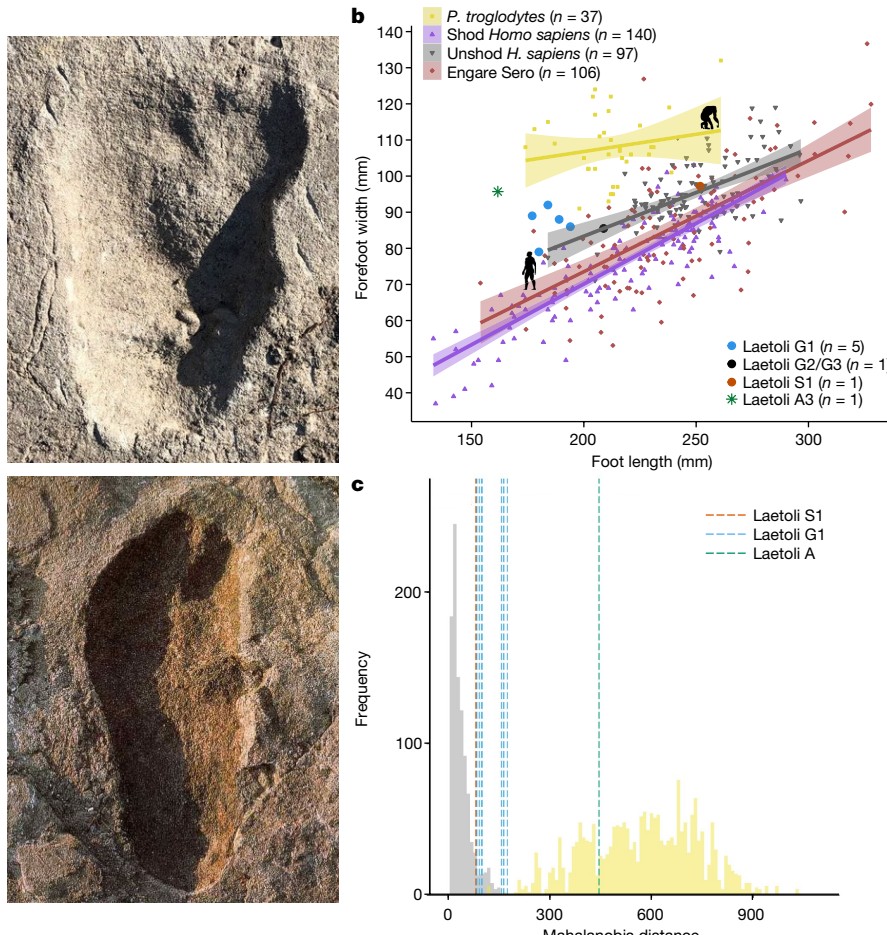

**Fig. 2 | Evidence against Laetoli A belonging to *A. afarensis*. a**, Images of Laetoli A3 (top) length matched to Laetoli G1 (bottom). G1 print is reproduced with permission from Science Photo Library. **b**, Plot comparing foot length to forefoot width in adult and juvenile humans and bipedal chimpanzees, the Pleistocene human footprints at Engare Sero, and the Laetoli trackways. Lines represent ordinary least-squares regression and shaded bands represent 95% confidence interval of the regression. Data were analysed from the total number of individual footprints indicated in the figure legend. Laetoli footprint markers are slightly enlarged for clarity. Data sources match

Extended Data Table 1. **c**, Histogram of Mahalanobis distances between the mean modern human footprint and the averages of two randomly drawn human footprints (grey; $n = 245$, resampled 1,000 times) and two randomly drawn bipedal chimpanzee footprints (yellow; $n = 45$, resampled 1,000 times). The blue and orange lines represent the distances of mean Laetoli G1 ($n = 5$, with 10 unique two-track combinations) and S1 ($n = 2$) two-track samples, respectively. All Laetoli G1 and S1 samples fall within the human distribution. The green line indicates the distance between the mean Laetoli A track ($n = 2$) and the human mean, falling squarely within the chimpanzee distribution.

an examination of foot ontogeny and intraspecific morphological variation that takes into account the mounting fossil evidence of locomotor (and presumably taxonomic) diversity among Pliocene hominins (for example, in ref. [33]).

Standing between 101 and 104 cm tall (from equations in Dingwall et al.[34]), the maker of the site A footprints was smaller in height than other Laetoli trackmakers, which ranged from 111–116 cm (site G1) to 161–168 cm (site S1)[35]. It is plausible that the site A tracks were made by a juvenile *A. afarensis*, but this hypothesis is undermined by a distinct footprint morphology from those at sites G or S.

The ratio of foot width and length follows a different ontogenetic trajectory in humans and chimpanzees; human feet are consistently narrower than chimpanzee feet (Fig. 2). Footprints made by unshod humans from both modern times and the Pleistocene are slightly wider than those made by shod humans in industrialized populations. The undistorted footprints from sites G and S fall within the human distribution. A3 is more chimpanzee-like in being wide compared with its length (Fig. 2a, Extended Data Fig. 2). In chimpanzees, this wider footprint shape is, in part, driven by the greater divergence of the hallux. We thus measured hallucial divergence as a ratio of the distance between the centre of the impression made by the first and second digits and the

length of the footprint. For this metric, humans and chimpanzees are clearly distinct. The best-defined site G footprints overlap the human distribution, whereas the A3 footprint does not—it possesses a slightly more divergent hallux than humans and site G, although not nearly as divergent as those of chimpanzees (Extended Data Fig. 6). This finding alone does not rule out a juvenile *A. afarensis*, given the foot from Dikika (Afar, Ethiopia) has a slightly more divergent and mobile hallux than its adult counterparts[36].

To explore other instructive traits, we compared proportional toe depth ratios as described by Raichlen and Gordon[37] (Supplementary Methods). The mean value for site A (−0.191) is distinct from Laetoli G1 and humans using a bent-hip bent-knee gait but overlaps the low end of variation in Laetoli S. Additionally, tracks A1–A3 evince a raised ridge of hardened ash between the heel and lateral forefoot. It is unclear whether this ridge is evidence of substrate shearing[38] or midfoot mobility—a characteristic absent from the Laetoli G and S footprints[39] (but see ref. [40]) and inconsistent with *A. afarensis* pedal remains[41].

Finally, we tested whether we could randomly sample footprints with internal topography similar to the various Laetoli footprints from those of humans or chimpanzees (following Hatala et al.[42]). Fig. 2 illustrates how the tracks from sites G and S can be encompassed in the

range of resampled unshod human footprint variation, whereas the average morphology of the A2 and A3 tracks is distinct from the footprints of habitually unshod humans and those at sites G and S. In fact, they fit comfortably within the resampled chimpanzee distribution, being as distinct as chimpanzee tracks are from the morphologies of barefoot human tracks. One possible explanation for such different footprint morphology is that the site A footprints were made by a cross-stepping *A. afarensis*. We tested this hypothesis by comparing the footprints of humans (n = 10) walking with their preferred gait and then cross-stepping. We found that normal and cross-stepping human footprints differ minimally and do not match in magnitude or direction the differences between the site G and S prints and the site A prints (Extended Data Fig. 7, Supplementary Information).

We therefore conclude that the site A footprints were made by a bipedal hominin with a distinct and presumably more primitive foot than *A. afarensis*. The gross shape of the foot is chimpanzee-like, with slight hallucial divergence and perhaps some midfoot mobility. However, the site A individual was walking bipedally with a narrow step width indicative of either a valgus knee, adducted hips, or both. This combination of foot morphology and gait kinematics inferred from the preserved footprints precludes them from having been made by *A. afarensis*.

Evidence is building for taxonomic diversity in hominins during the Pliocene[43,44], including at Laetoli[45–47], but these hominins did not walk with morphologically identical feet[48]. For example, the BRT-VP-2/73 foot from the 3.4 Ma site of Woranso-Mille, Ethiopia demonstrates that at least two different foot morphs co-existed in the Afar Depression during the Pliocene[33]. We suggest that footprint evidence for hominin locomotor diversity is similarly present at Laetoli, Tanzania—and has been since the discovery of the site A trackway in the 1970s.

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

## Methods

### Wild bear behavioural data

Wild black bear behaviour was quantified using video data recorded by B.K. over the course of several years at his ongoing field site in Lyme, New Hampshire, USA. Video data captured bears of different ages (cubs, adolescents and adults). Bears were present on screen for a total of 50 h 55 min 18 s. For each terrestrial bipedal incident, the length of the event, the approximate age of the bear and the number of steps were recorded. Additionally, steps were evaluated on whether they were completed independently, or the individuals used other environmental objects for balance.

### Comparative kinematic data

Comparative kinematic data were collected on three species: *U. americanus*, *P. troglodytes*, and *Homo sapiens*. For bears and chimpanzees, the sample size included all available individuals housed at each location. For the human sample size information see below. Randomization was not relevant to our study as we were interested in measuring footprint characteristics from whole sample populations, as opposed to comparisons within those populations. Blinding was not relevant to the data collected on the non-human comparative species (for example, bears and chimpanzees) nor to the data collection on fossilized footprints. The human participants were unaware of the site A tracks at Laetoli and therefore had no knowledge of how the data obtained from their footprints would be used in this study.

### *Ursus americanus*

Data were collected on four juvenile semi-wild *U. americanus* ($n = 3$ male, 1 female), whose feet were within 10% of the length (average foot length = 145.7 mm) of the recorded footprints of Laetoli site A (average foot length = 161.7 mm). These orphaned, approximately 20-kg bears were located at the Kilham Bear Center (Lyme, NH), awaiting reintroduction to the wild. This study examined the bears between the ages of 5–8 months old. Our protocol was reviewed and approved by the Institutional Animal Care and Use Committee (IACUC) of Dartmouth College. The bears were enticed to independently walk bipedally through a constructed mud trackway for either an applesauce or maple syrup reward (Extended Data Fig. 3). Measurements were collected on the footprints, including foot length, heel width, forefoot width, step length and stride width using the definitions from Tuttle[4]. For a subset of footprints ($n = 5$), the width of the impression for the 1st, 2nd, 4th and 5th digits were measured. The presence or absence of claw impressions was also documented.

### *Pan troglodytes*

Data for extant chimpanzees were extracted from three sources to collect all the relevant gait metrics. Two published datasets examined the same two subadult individuals housed at Stony Brook University. The third set were recorded on semi-wild individuals ($n = 46$), using a plantar pressure mat at the Ngamba Island Chimpanzee Sanctuary (Entebbe, Uganda). While this third data set increases sample size and captures intraspecific variation, we recognize that plantar pressure data do not always align perfectly with footprints made in a deformable substrate[20].

**Stride width data and step length comparisons.** Chimpanzee stride width data were taken from Thompson et al.[28] on two subadult male chimpanzees ($7.0 \pm 0.1$ years of age; $34.8 \pm 1.2$ kg) and were supplemented with step length data for the same steps. Three-dimensional kinematic methods and step width calculation have been described previously[28]. Step length was calculated as the distance between left and right calcaneus markers in the sagittal plane during consecutive hind limb midstance periods. Chimpanzee step lengths are typically asymmetric, so step length was averaged over the two consecutive steps which defined the stride.

**Forefoot width, heel width and foot length comparisons.** Footprint dimensions and stride length data were recorded on the same two subadult male chimpanzees as above, though at a slightly younger age (6.5 and 6.9 years of age, 30.7 and 27.8 kg, respectively). The experimental design is described in detail elsewhere[42]. In brief, chimpanzees traversed a runway, at the centre of which was a pressure mat (RSScan International) and a container of hydrated sediment in which the chimpanzees could produce footprints. This sediment was taken directly from a layer that preserves 1.5 Ma hominin footprints near Ileret, Kenya[50]. Laterally positioned video cameras were used to record the chimpanzees as they walked along this trackway and produced footprints. Two digitization softwares, MaxTRAQ Lite+ (v. 2.4.0.3) (Innovisions Systems) and ImageJ v.1.47[51], were used to quantify various aspects of their gaits, including stride length. Tape measures and digital callipers were used to directly measure the external dimensions of each chimpanzee's feet. Scaled photographs were taken of the footprints produced in each trial, and these were later measured using ImageJ software.

**Forefoot width, width of digits 1 and 2, divergence ratio, and foot length comparisons.** Data were collected by E.J.M. at the Ngamba Island Chimpanzee sanctuary (Entebbe, Uganda) managed by the Chimpanzee Sanctuary and Wildlife Conservation Trust (CSWCT) using procedures approved by the Dartmouth College IACUC. The Tekscan plantar pressure mat (PPM) was positioned within a walkway connecting the overnight enclosure to the open forest habitat, near a gate and underneath a solid cross section to help prevent individuals from jumping over the mat using the ceiling bars. This location was determined using the expertise of the sanctuary keepers. The animals were first introduced to a mat shell that was lacking the internal sensors, to habituate them to the novel stimulus. All subsequent data were collected using both the empty and real PPMs, positioned such that they covered the entire width of the walkway to force individuals to walk across one of the two mats. Both mats were covered with thin green sacks to help disguise them from the chimpanzees and facilitate faster removal if necessary. It was determined that the southeast facing direction was the preferred path for the chimpanzees and the sensor-containing PPM was always positioned there from the second collection onwards. Data were collected twice a day; once in the morning (between 06:45 and 08:00) when the chimpanzees were headed to the forest for their first feeding, and once at 18:00, when the chimpanzees were headed into the overnight enclosure to sleep and receive their last feeding. Data were collected on 46 adult chimpanzees (18 male, 28 female, ages 12–36 years). A subset of 54 dynamic pressure records was analysed. Using the associated Tekscan PPM software, Footmat Research (v. 7.10), the pressure recordings were analysed to determine foot length, forefoot width, the width digits 1 and 2, and the linear distance between the centre of digits 1 and 2. The divergence ratio was calculated by dividing the distance between digits 1 and 2 by the individual's foot length.

### *Homo sapiens*

Data were extracted from previous studies of two modern human populations in order to collect all the relevant foot, footprint and gait metrics.

**Stride width and step length comparisons.** Data were taken on 654 participants, recruited through the Living Laboratory at the Boston Museum of Science[19]. Sample size was determined by museum visitor traffic and willingness to participate in a scientific study. In brief, this dataset included 73 children between the ages of 2 and 7 years old (29 female and 44 male) and 581 individuals (366 female and 215 male) between the ages of 8 and 80 years. A pressure-sensitive gait carpet (6.1 m long × 0.89 m wide) with a spatial resolution of 1.27 cm and collecting data at 120 Hz (GAITRite) was used to collect stride length and stride

width. For a subset of 33 adults, additional data were collected using a Tekscan PPM and analysed with FootMat Research (v. 7.10) to calculate foot length, the width of digits 1 and 2, and the linear distance between the centre of digits 1 and 2. These measurements were used to calculate a divergence ratio as described above in '*P. troglodytes*'.

**Forefoot width, heel width and foot length comparisons.** Footprint dimensions and stride length data for 29 Daasanach adults (15 male and 14 female, ages 18–47) and 12 children (10 male and 2 female, ages 4–15), who live near the town of Ileret, Kenya and grew up either habitually unshod or minimally shod were taken from Hatala et al.[20,21]. Details of the experimental protocol largely mirrored the procedures described above. In brief, subjects generated footprints while walking through a rehydrated sample of the same sediments that preserve 1.5 Ma hominin tracks near Ileret. Video cameras were used to record subjects as they produced footprints, and two digitization software packages (Max-TRAQ Lite+ v. 2.4.0.3 and ImageJ v.1.47) were used to measure stride lengths and other kinematic variables. The external dimensions of subjects' feet were directly measured with tape measures and digital callipers. Scaled photographs of the footprints produced in each trial were measured using ImageJ.

**Human cross-stepping footprint experiments.** Experiments were carried out by K.G.H. and E.M.W.-H. to investigate whether and how cross-stepping kinematics influence the perimeter dimensions and internal topologies of an individual's footprints. We could thereby evaluate whether the size and shape of the Laetoli site A tracks could have been generated by a hominin with feet similar to those who left tracks at sites G and S, but while cross-stepping. Detailed methods are provided in Supplementary Information. In brief, ten adult subjects (including six female, three male, and one non-binary between 19 and 52 years old) each completed ten trials in which they produced tracks in sedimentary conditions meant to mimic those at Laetoli[37,52]. Sample size was determined by availability and willingness to participate in the study. Five trials were completed with a normal, self-selected walking gait and another five were completed with a cross-stepping gait, as inferred for the Laetoli site A trackmaker. In each trial, a focal footprint was selected, measured in situ, and photographed (25–30 photos per footprint). The lengths and widths of the steps bracketing the track were also measured. Photographs were used to generate 3D models of the tracks using Agisoft Metashape software (v.1.7.3), and average normal and cross-stepping tracks were generated for each subject using DigTrace Pro (v.1.8.1)[53]. Lengths and widths of these averaged tracks were measured using Geomagic Wrap (v. 2021.0.0) (3D Systems). Regional depths were measured and evaluated using the same methods described below ('Comparative analyses of Laetoli footprint shapes'). Within-subject comparisons enabled us to understand how cross-stepping influenced the dimensions of the perimeter and the internal topology of a subject's footprints.

## Fossil footprint data and analysis
Comparative metrics were quantified from a set of modern human footprints from the Late Pleistocene at Engare Sero, Tanzania. These footprints are an important comparison with the Laetoli footprints, as they were generated in a similar circumstance (footprints in volcanic ash) and represent an early population of unshod modern humans.

A scaled 3D orthophoto of the Engare Sero site was created via photogrammetry by B.Z. and C.L.-P. to visualize the distribution of footprint trackways across the entire site using Agisoft Photoscan (now Agisoft Metashape v. 1.4.4). The model was created from hundreds of photos originally taken by the Smithsonian 3D Digitization Program in 2010. Measurements that were defined in Tuttle[4], were taken from the fossil tracks using the software, ImageJ (v. 1.49), and included foot length, forefoot width, heel width and step width of each footprint at Engare Sero. In some cases, partial footprints were included for measurement,

as long as they included the requisite landmarks for those measurements. Overall, data were collected from 151 footprints at the Engare Sero site. Of the 151 footprints, 61 footprints were considered partial footprints and 90 footprints were considered complete footprints. From these, 67 step length and stride width measurements and 105 heel width and ball width measurements were included in our analyses.

All measurements for Laetoli trackways G and S were obtained from published sources[4,32,42]. Box and whisker plots and bivariate graphs (using ggplot2[54]) were generated using R (v. 3.6.1), while the table and pie chart were generated using Microsoft Excel (v. 2102).

## Comparative analyses of Laetoli footprint shapes
Comparative analyses followed methods similar to resampling analyses published previously[42]. In brief, the human comparative sample included 245 footprints produced by 29 adult and 12 juvenile habitually unshod Daasanach individuals traveling at walking speeds. The chimpanzee comparative sample included 45 footprints produced by two individuals walking bipedally. Laetoli samples included only the best-preserved tracks from each site, leaving samples of five footprints from site G that were described by their original excavators as free from taphonomic damage that would obscure track topology (G1-25, G1-27, G1-33, G1-34 and G1-35), and two from site S (L8-S1-2 and L8-S1-4). For site A, we included tracks A2 and A3, as these were the only two for which we were relatively confident in identifying regions of interest across the entire track. Larger sample sizes would be desirable, but we did not want to sacrifice data quality for quantity by including tracks that were overprinted or that did not appear to represent complete foot anatomy. We did not rely on parametric statistical tests for which larger sample sizes would be a necessity, and instead used an analytical approach that could handle smaller sets of observations (see below).

For each experimental and fossil footprint, 3D models were constructed using photogrammetry, through a variety of methods described here for Laetoli site A and elsewhere by the authors for other samples[20,21,35,42]. Using Geomagic Wrap (v. 2021.0.0) (3D Systems), a best-fit plane was fit to the undisturbed substrate surrounding each track, and this was fixed to the $xy$ plane in world coordinate space. In this orientation, depths of the footprint were measured in the regions of the medial and lateral heel, medial and lateral midfoot, and all five metatarsal heads and toes. Raw depth measurements were normalized, within each footprint, to a scale of 0 to 1 in order to compare the topologies of footprints that may vary in depth. However, a Wilcoxon signed-rank test showed that, overall, human and Laetoli track samples did not differ significantly in their depths ($P = 0.08$). Within-subject means of the 14 normalized depth measurements were calculated, and a between-subject covariance matrix was created using the subject averages for normalized depths at each of the 14 measured regions. An overall 'human mean footprint' was also computed by averaging the within-subject mean normalized depths, and this represented a measure of central tendency as described below.

To represent the range of observed variation in human footprint topography, for 1,000 iterations we randomly sampled a human subject and drew a sample of two of their footprints. We then averaged the normalized depths of those two footprints and computed the Mahalanobis distance (using the between-subject covariance matrix) between this track and the mean of all other subjects' footprints. Also, for 1,000 iterations we selected a random chimpanzee subject, drew a random sample of two of their footprints and computed the Mahalanobis distance between the average of those tracks and the overall mean human footprint. For the Laetoli tracks, site A and site S samples only included two tracks, so these were simply averaged and the Mahalanobis distance was calculated between each averaged track and the mean human footprint. For Laetoli site G, all possible two-track combinations (ten) were drawn from the sample described above, and the Mahalanobis distance was calculated between the averaged track from each combination and the human mean. In all cases, we calculated multivariate distances

using the human between-subject covariance matrix (that is, treating the chimpanzee and fossil tracks as if they came from different human subjects). All analyses described above, and the histogram displaying multivariate distances (Fig. 2), were generated using R (v. 3.6.1), with custom scripts and functions from the dplyr[55], ggplot2[54] and reshape2[56] packages.

## Photogrammetry

While casts of the site A bipedal footprints existed at one point, all our attempts to locate them (see Acknowledgements for a complete list) were unsuccessful. Prior to our fieldwork at Laetoli in 2019, we modelled the original trackway photogrammetrically using extant photography from the site. Original photography of Laetoli site A was taken by J.R. We obtained his photographs of trackway A through Science Photo Library. All photographs were taken with a Nikon F2 on 35-mm Kodachrome slide film. Digital scans from these slides were used to produce a 3D model of the Laetoli A footprints. Unfortunately, since the images were taken in 1977, they were not recorded with modern photogrammetry processing in mind. Several features of the digitized images limit successful and accurate construction of a 3D model. First, there are only four images of the footprints. One of these images has noticeably different exposure settings that caused significant alignment problems during processing, and thus was excluded. All images were shot at oblique angles, from a relatively narrow range of camera positions. A yellow string defining the site grid lies over one of the footprints, obscuring part of it, and casting a shadow. The images were all taken relatively early in the day, so there are shadows within each footprint that create strong contrasts. The slides were digitized at 4,000 dpi, but they were not scanned with specialized equipment to guarantee geometric accuracy, and this potentially introduced more sources of distortion.

However, despite the limitations of the images, it was possible to extract 3D data for the Laetoli A footprints. All processing was done in Agisoft Photoscan Pro (v. 1.7.1). The standard processing steps (align photos, build dense cloud, build mesh, build texture) were run to produce a 3D model, though the process had to be done iteratively to remove noise, ensure accurate alignment of the photos, scale the model appropriately using published measurements, add manual tie points, and refine the model. A DEM (digital elevation model) and orthophotograph were exported for further visualization and analysis in ArcGIS (v. 10.6.1). The 3D model was also exported to Autodesk Meshmixer (v. 3.5.474) to create a 'watertight' 3D volume that could be 3D printed for further visualization (1977 model is hosted on Morphosource, ID: 000390119). Photogrammetric reconstruction was validated using published measurements of the footprints. It is important to note however, that there were no published measurements for the depths of the footprints and that the internal anatomy of this reconstruction is potentially misleading because of the incomplete excavation of the footprints[2,8,12].

A second, more accurate reconstruction was done using photogrammetry from the re-excavated site A bipedal trackway using 57 images taken in June 2019. The images were captured in a systematic manner using a Nikon D7000 camera and Nikon DX AF-S Nikkor 18–105 mm lens. All photos were taken by hand, from an eye level, while walking a series of transects, across the area of interest. Spacing between shots was kept low to ensure a minimum of approximately 65% overlap between adjacent images. All processing was done using Agisoft Photoscan Pro/Metashape Pro (v. 1.7.1). Standard processing steps (for example, as described[15,51]) were taken to create a 3D model of the A trail. This included photo alignment, manual editing of the sparse cloud to remove points with high 'reprojection uncertainty', building a dense cloud, building a mesh, refining the mesh, then building a texture. During processing, images were checked for sharpness using the 'image quality' tool and any images with significantly lower quality were removed. The model was scaled to the real-world using scale bars placed across the region of interest. Finally, an orthophotograph as

well as a DEM (digital elevation model) were exported as geotiffs into ArcGIS in an arbitrary local coordinate system for further analysis (2019 model hosted on Morphosource, ID: 000390114).

To generate contour maps, two approaches were used. First, starting with the raw stereolithography scans (.stl file format), Ultimaker Cura software (v.4.8.0) was used to rotate the raw scans and align them with x and y axes. This was a manual process. These rotations were exported to binary-format .stl files. The rotated files were then run through an R script using R version 4.0.3. The R script uses the tidyverse and rgl libraries to load the .stl files into R-friendly dataframes and plot them as contours using ggplot's geom_contour function. The script is available through GitHub.

Using a second approach, the .stl files were brought into Cloud Compare (v. 2.11.3) to check model orientation. If necessary, models were reoriented to allow the local ground surface to be level using the "level" tool, and then the files were exported. The correctly oriented model was imported into SAGA GIS using the import stereo lithograph file (STL) tool. This tool converts the .stl directly to a DEM raster. The rasters were checked in SAGA and a hillshade generated with the analytic hillshading function using the standard sun position setting of 315° azimuth and 45° height. Both the DEM and hillshade were then exported as geotiffs. These geotiffs were imported into ArcGIS for visualization. The DEM was colored using a red-blue colour ramp to indicate relative depth and this was layered onto the hillshade raster using the NAGI fusion method[57] (Extended Data Fig. 8). Cloud compare was used to quantify erosive alterations to the site A footprints from 1977 to 2019.

## 3-D surface scanning

Three-dimensional surface scans of Laetoli A and plaster casts of bear prints were collected using a Creaform Go!Scan 50.

## Reporting summary

Further information on research design is available in the Nature Research Reporting Summary linked to this paper.

## Data availability

Previously published data were obtained from refs. [4,19–21,28,32,42]. All other data supporting the findings of this study are available within the paper and its supplementary information files. The photogrammetric reconstruction of the Laetoli A trackway based on three original photographs from the 1977 expedition is available on Morphosource (https://www.morphosource.org; ID: 000390119). The photogrammetric reconstruction of Laetoli A trackway using 57 photographs taken of the re-excavated Laetoli site A footprints in 2019 is available on Morphosource (ID: 000390114). Source data are provided with this paper.

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

**Acknowledgements** We thank R. Tuttle for his mentorship and inspiration; M. Larkin for her help collecting bear footprint data; R. Leakey, B. Wood, H. de Lumley, T. White, R. Crompton, R. Clarke, W. Harcourt-Smith, C. Stringer, H. Bonney and the London Natural History Museum, and J. Kibii and the National Museum of Kenya for assistance as we attempted to track down photographs and casts of the Laetoli A footprints; J. Sappington for photographic support; and V. Rossi and A. Metallo for help with 3D visualization; the Director General of Tanzania Commission for Science and Technology (COSTECH) for granting our team the COSTECH

Research Permits (2019-370- NA-2019-2016); F. Manongi, J. Mwankunda and M. Mwambungu for allowing us to camp and work at Laetoli. Funding for this research was provided by the National Geographic (8748-10), Leakey Foundation (71483-001), National Science Foundation (BCS-1128170, DGE-080163, BCE-1730822, GRFP (No. 1840344), the Smithsonian Institution's Human Origins Program, the Evolving Earth Foundation, the Explorers Club, and the Claire Garber Goodman Fund-Dartmouth College.

**Author contributions** E.J.M. contributed to study generation and design, data collection on bears and chimpanzees and analyses, and wrote the manuscript. K.G.H. contributed to data collection on humans and chimpanzees and analyses, and assisted with the manuscript. C.M. contributed to data collection during fieldwork at Laetoli and analyses. J.C. and A.C.H. contributed the photogrammetry analysis. J.A., A.C.H. and S.G. developed the contour maps of the footprints. A.S.D., K.F., L.D.F., J.G., E.G., S.K., B.M., A.P., S.R., R.T. and C.M.M. contributed to data collection during fieldwork at Laetoli. N.J.D. contributed to study design and assisted with the manuscript. S.V.G., J.M.D., K.G.H. and E.M.W.-H. contributed data collection on humans. J.M.D. also contributed to data collection during fieldwork at Laetoli, study design, and assisted with the manuscript. C.J. contributed to data collection on bears and analyses. B.K. and P.K. contributed to access to wild bears and data collection. C.L-P. and B.Z. contributed to access and data collection on the Engare Sero footprints. E.K. contributed to analyses of the Engare Sero footprints. J.R. contributed original photography of Laetoli Site A. N.E.T. contributed to data collection on chimpanzees. All authors contributed editorial comments to the manuscript.

**Competing interests** The authors declare no competing interests.

**Additional information**
**Correspondence and requests for materials** should be addressed to Ellison J. McNutt.

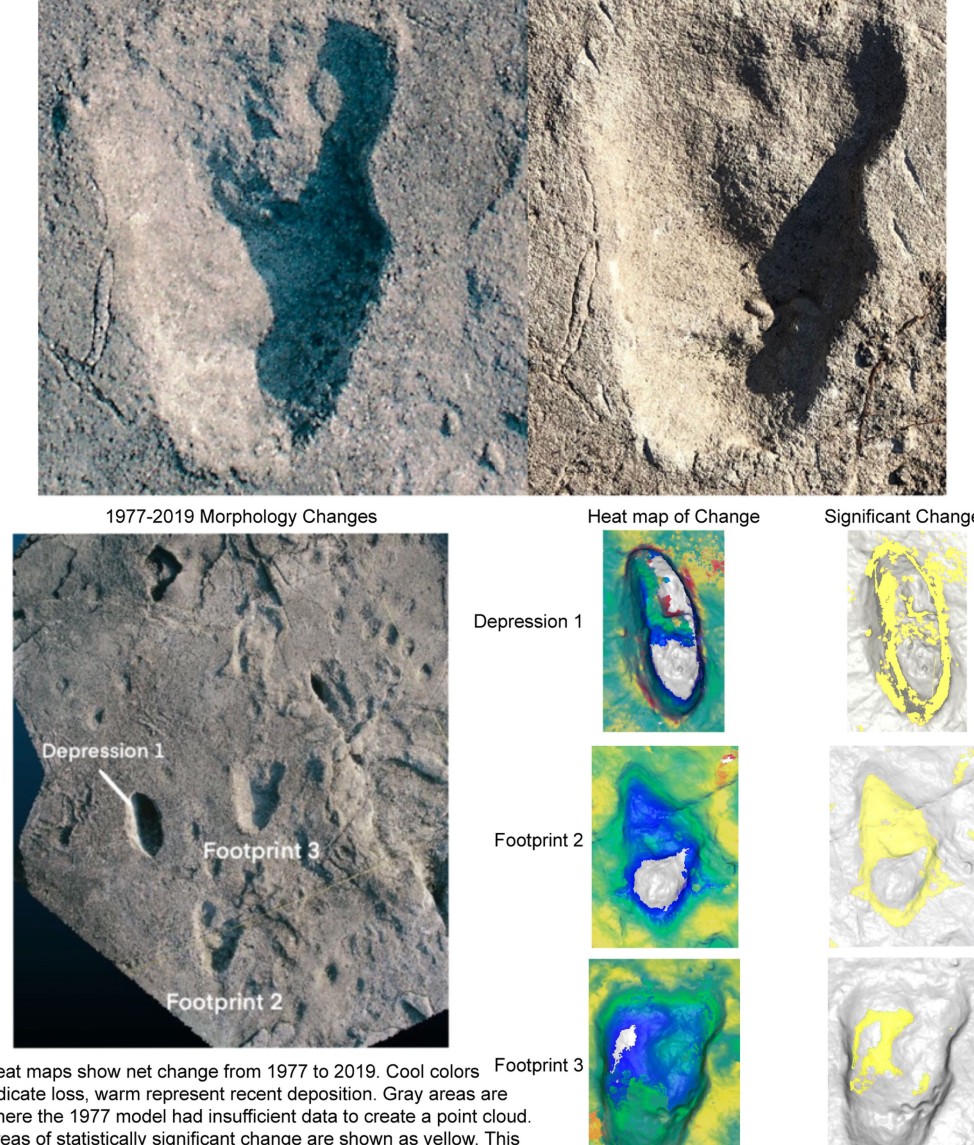

1977-2019 Morphology Changes

Heat map of Change    Significant Change

Depression 1

Footprint 2

Heat maps show net change from 1977 to 2019. Cool colors    Footprint 3
indicate loss, warm represent recent deposition. Gray areas are
where the 1977 model had insufficient data to create a point cloud.
Areas of statistically significant change are shown as yellow. This
change is in excess of any model construction or alignment errors.

**Extended Data Fig. 1 | Laetoli print A3 and erosion.** Photographs were taken by J. Reader (left: 1977) and J. DeSilva (right: 2019) at similar overhead angles and times of day (see similarities in shadows cast across the print). Notice the removed matrix infill in the hallucial impression and the presence of the previously unseen second digit impression in 2019 image. Below: comparisons of the photogrammetric meshes of a pothole and footprints A2 and A3 using Cloud Compare[15]. The impact of erosion on the morphology of the footprints was assessed by quantifying changes to an oval depression located west of A3. Notice that significant changes occurred around the rim of the depression, as would be expected through erosion, that are absent around the rims of the A2 and A3. Instead, significant differences between A2 and A3 are located internally and are a result of a more thorough excavation of the prints.

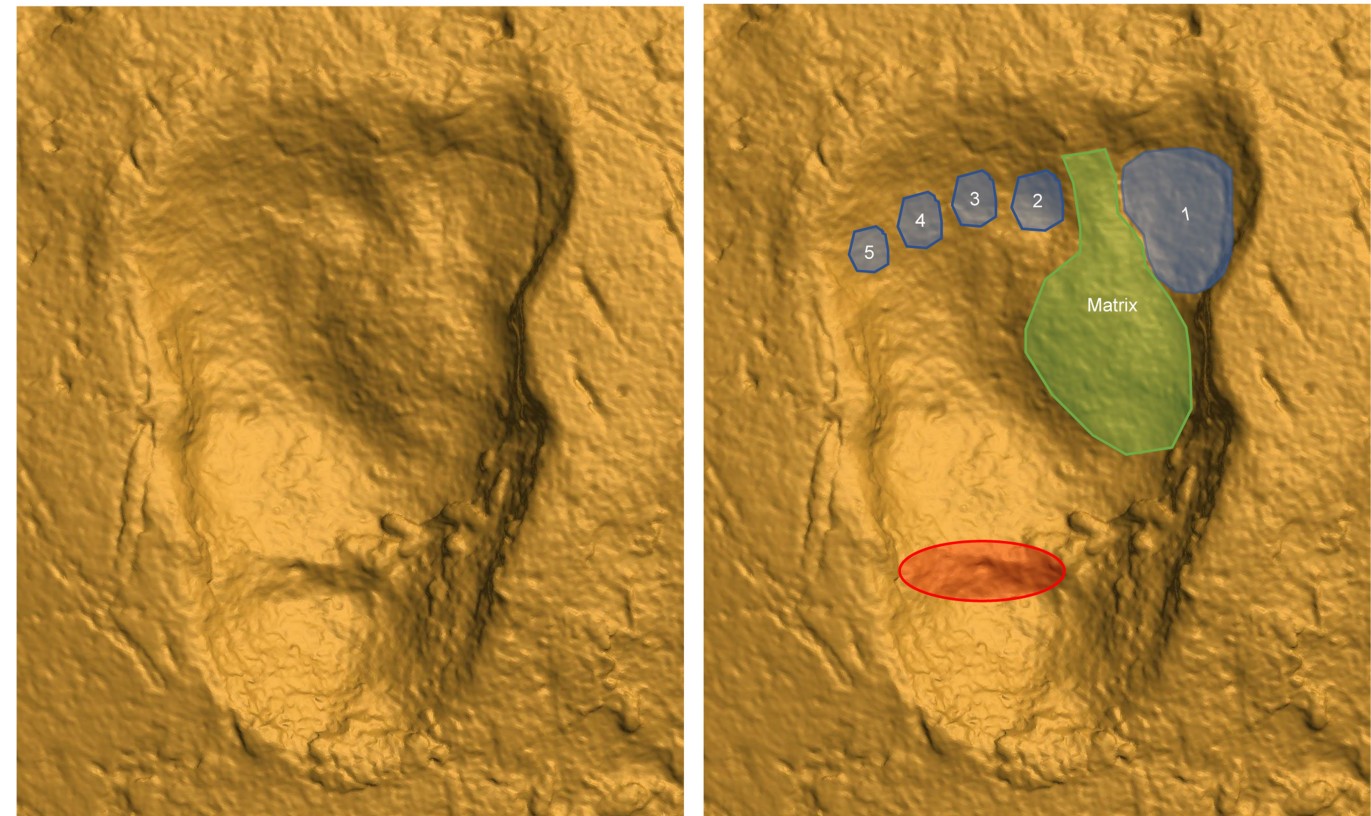

**Extended Data Fig. 2 | Details of the best preserved Laetoli A hominin footprint (A3).** Left image shows original 3D scan. Right image highlights the proposed impressions for the toes (blue circles) and matrix infill (green), as well as the potential evidence for midfoot mobility (red).

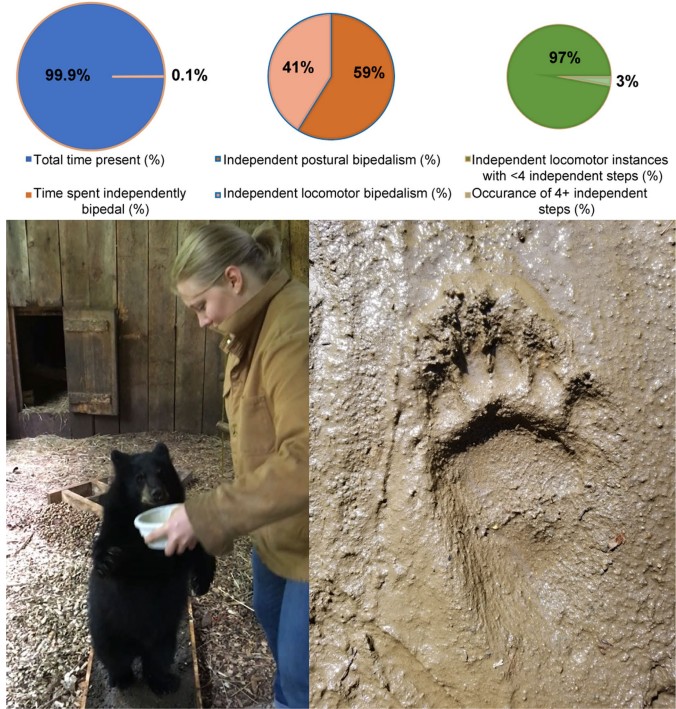

■ Total time present (%)　　■ Independent postural bipedalism (%)　　■ Independent locomotor instances with <4 independent steps (%)

■ Time spent independently bipedal (%)　　■ Independent locomotor bipedalism (%)　　■ Occurance of 4+ independent steps (%)

**Extended Data Fig. 3 | Incidence of bipedalism in *Ursus americanus* and Examples of kinematic data collection.** (top) Pie charts showing the frequency of bipedal behaviors in wild *Ursus americanus*. The blue chart represents the time spent independently bipedal (locomotor or postural) out of the total 50.9 h in which bear behavior was observed. The orange chart represents the breakdown of time spent independently bipedal into its postural and locomotor (i.e., bears took one or more steps) components. The green chart represents the frequency of occurrences where bears walked 4 or more steps reflecting a similar circumstance to Laetoli trackway A. (bottom left) Juvenile female walks bipedally, unassisted through mud trackway. (bottom right) Example left footprint from one of the juvenile males.

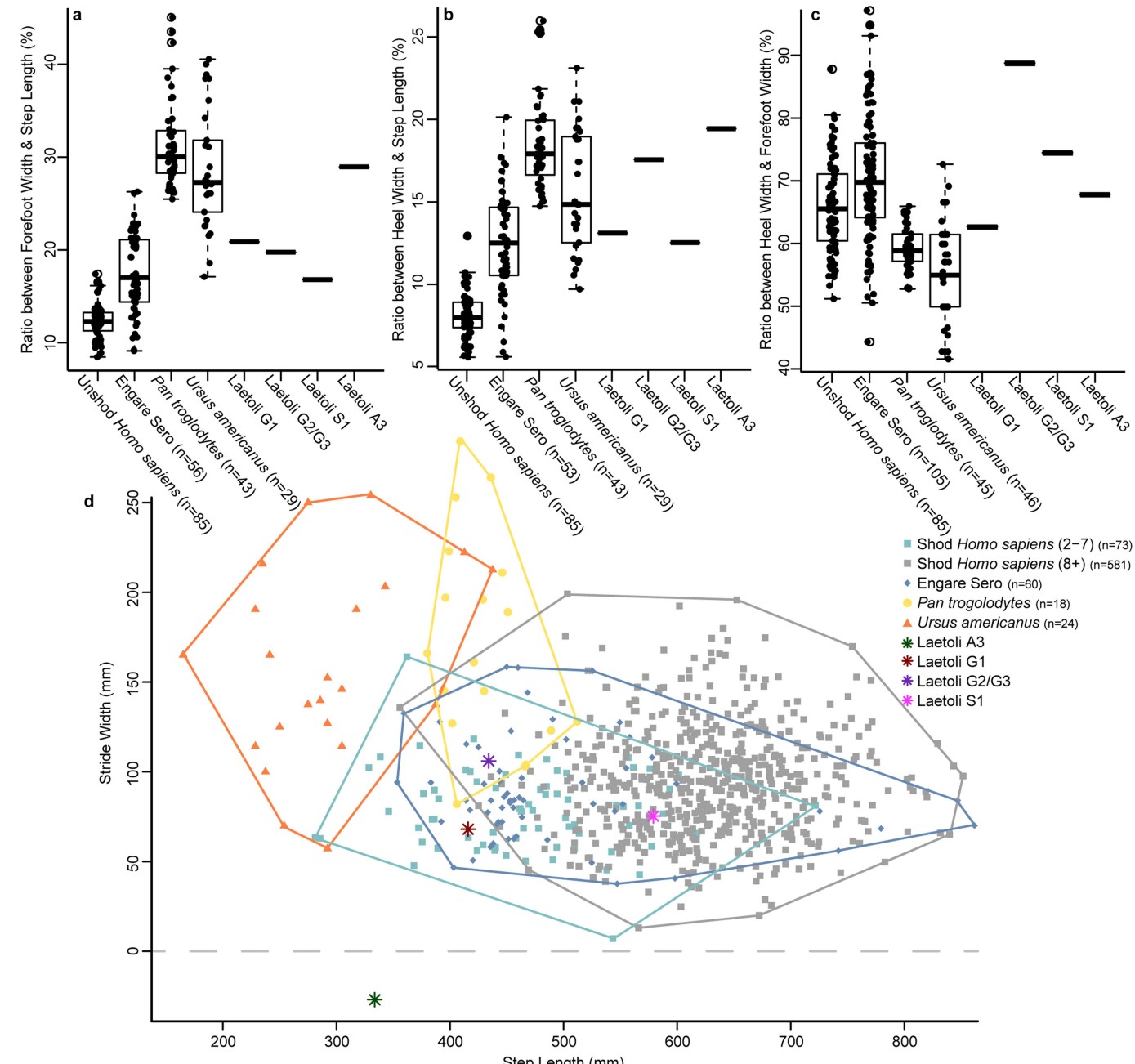

**Extended Data Fig. 4 | Foot and gait comparisons across all comparative species.** (a) Forefoot width to step length; (b) heel width to step length; (c) heel width to forefoot width; and (d) stride width to step length. Notice that for foot proportions (c) and stride width (d), Laetoli A is unlike the tracks produced by bears. In (d) the negative value in Laetoli A represents the fact that the track is demonstrating cross-stepping. (a-c) Boxplot represents median (center line), upper and lower quartiles (box limits), range (whiskers), and outliers (points) and individual footprints sample sizes for each species are indicated in the figure panels. (a-d) In all plots, n = 1 for all Laetoli trackways, and chimpanzees were traveling bipedally. Data sources match Extended Data Table 1.

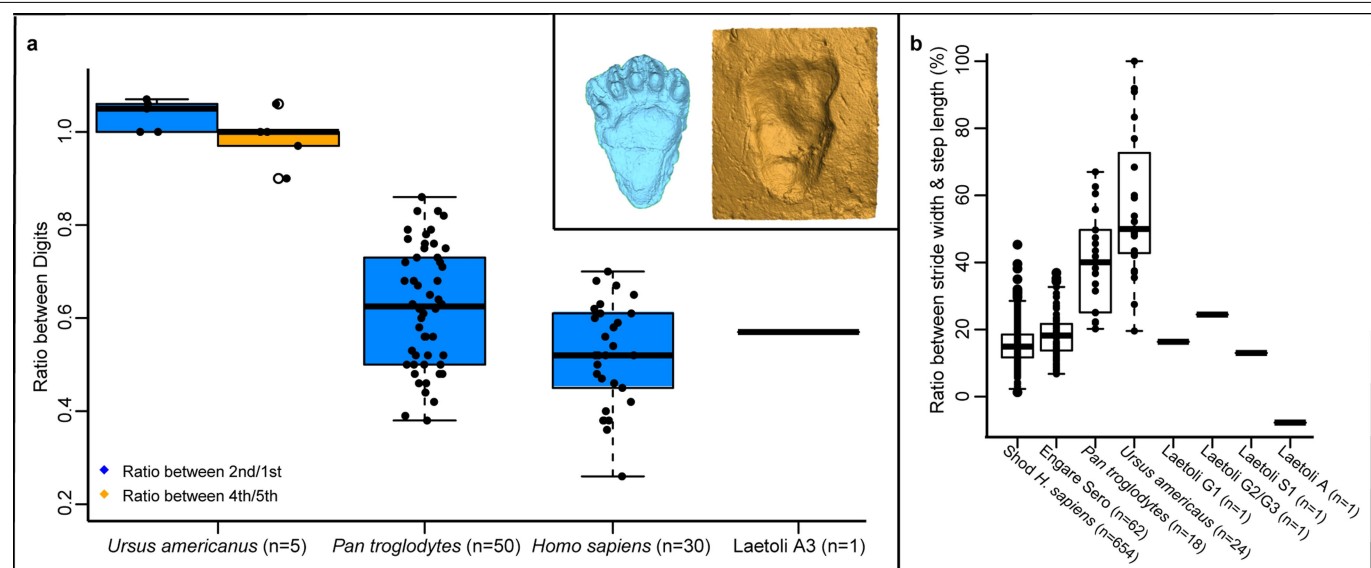

**Extended Data Fig. 5 | Evidence challenging ursid hypothesis.** (a) Ratio between toe impression dimensions across footprints from the comparative species and Laetoli A3. Values in blue=the ratio between the width of the 2nd digit compared to the hallux. Values in orange=ratio between the width of the 4th digit compared to the 5th in *Ursus americanus*. (Insert image) Comparison between 3-D scan of right ursid footprint (blue) and A3 (orange). Note the large size of the 5th digit impression in ursids but overall shape difference between the ursid track and A3. (b) Ratio of stride width to step length across the different species, including the Laetoli bipedal trackways. Boxplot definitions are as in Extended Data Fig. 4 and individual footprint sample sizes are indicated in the figure panels. In 5a, the chimpanzees were traveling quadrupedally; in 5b, chimpanzees were moving bipedally. Data sources match Extended Data Table 1.

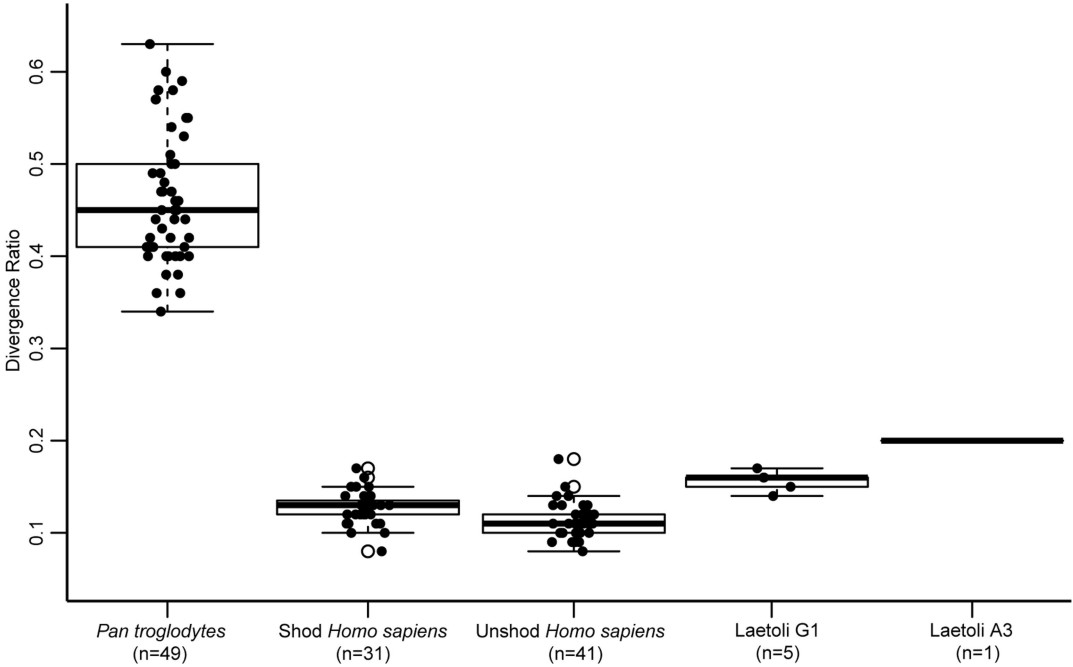

**Extended Data Fig. 6 | Boxplots showing the divergence ratio between the first digit and 2nd digit across the comparative species and two Laetoli trackways.** Divergence ratio was measured by dividing the linear distance between the midpoint of digits one and two by the foot length. Chimpanzees were traveling quadrupedally and data were obtained from plantar pressure impressions. Boxplot definitions are as in Extended Data Fig. 4 and individual footprint sample sizes are indicated in the figure panel. Data sources match Extended Data Table 1. We attempted to measure hallucial divergence following Bennett et al[50]. but were stymied by matrix obscuring the deepest region of the ball of the foot.

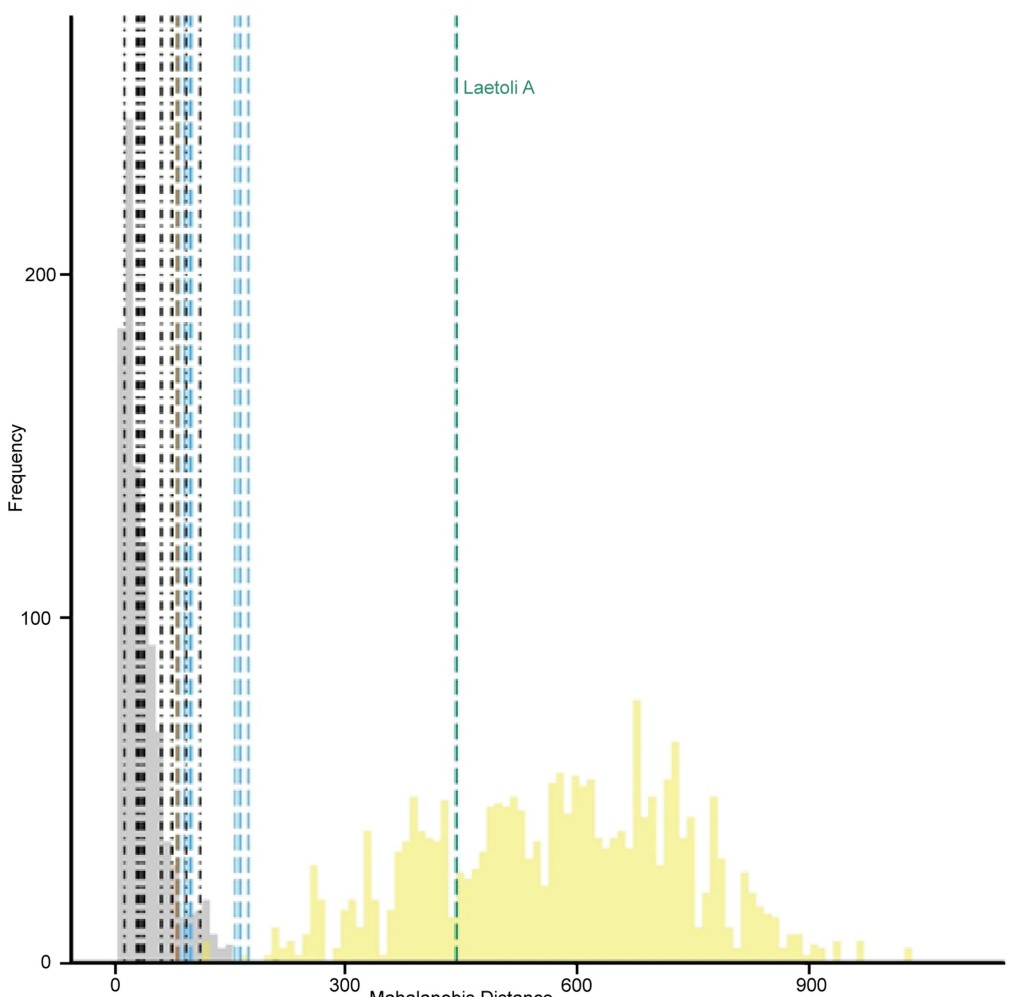

**Extended Data Fig. 7 | Histogram of Mahalanobis distances between the mean unshod human footprint and resampled unshod human footprints (gray) and chimpanzee footprints (yellow).** As in Figure 3c, blue, orange, and green dashed lines represent samples from Laetoli G1, S1, and A, respectively. Only Laetoli A is labeled, for clarity. Sample sizes for these samples as in Fig. 2c as well. The black dotted-dashed lines have been added to represent the average cross-stepping footprints produced by 10 adult habitually shod humans. All fall squarely within the distribution of unshod human footprints (probabilities of sampling tracks like these range from 0.20 to 0.94), and a great distance apart from the Laetoli A sample (green). Human cross-stepping footprints tended to be slightly closer to the human mean than the Laetoli S1 and G1 samples, but their distribution does overlap with the Laetoli S1 sample and with some of the Laetoli G1 samples. Cross-stepping footprints fell, on average, a Mahalanobis distance of 27.2 farther from the human mean than their "normal" walking counterpart.

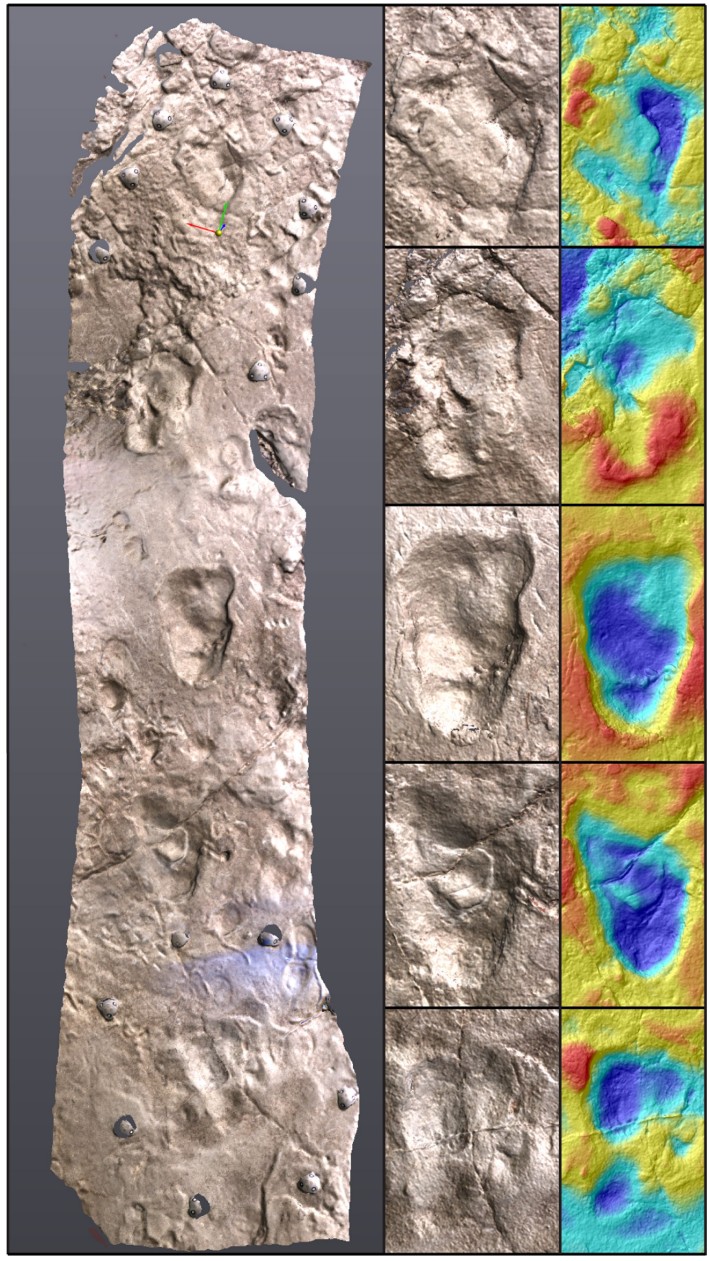

**Extended Data Fig. 8 | 3D scans of Laetoli A footprints and their contours.** Left panel: Complete scan of Laetoli A trackway with A1 at the bottom and A5 at the top. Right side: Zoomed in images of the individual footprints and their corresponding contour images.

## Extended Data Table 1 | Summary of average bipedal footprint metric data

| Species | Avg. Foot Length (mm) | Avg. Forefoot Width (mm) | Avg. Heel Width (mm) | Avg. Step Length (mm) | Avg. Stride Width (mm) | Heel/Forefoot Ratio (%) | Heel/Step Length Ratio (%) | Forefoot/Step Length Ratio (%) | Stride Width/Step Length (%) |
|---|---|---|---|---|---|---|---|---|---|
| Shod *Homo sapiens* (2-7)[*] | - | - | - | 469.2 (83.8) | 78.2 (22.5) | - | - | - | 16.7% |
| Shod *Homo sapiens* (7+)[*] | - | - | - | 628.7 (73.7) | 94.5 (30.6) | - | - | - | 15.0% |
| Cross-Stepping Shod *Homo sapiens* | 263.1 (13.1) | 111.7 (7.2) | 80.4 (4.2) | 537.4 (45.6) | -55.9 (59.5) | 72.0% | 15.0% | 20.8% | -10.4% |
| Unshod *Homo sapiens*[†] | 250.3 (21.4) | 95.5 (9.0) | 63.3 (9.8) | 803.0 (111.9) | - | 66.3% | 7.9% | 11.9% | - |
| Pleistocene *Homo sapiens* (Engare Sero) | 239.1 (34.8) | 85.5 (15.6) | 60.2 (10.8) | 512.5 (149.6) | 89.6 (28.3) | 70.4% | 11.7% | 16.7% | 17.5 % |
| *Pan troglodytes* | 203.3 (9.8)[‡] | 112.6 (4.6)[‡] | 66.9 (4.2)[‡] | 430.1 (35.9)[§] | 171.7 (57.7)[§] | 59.4% | 15.5% | 26.2% | 39.9% |
| *Ursus americanus* | 145.7 (15.6) | 78.9 (9.6) | 43.3 (7.0) | 292.8 (80.6) | 156.9 (48.1) | 54.8% | 14.8% | 27.0% | 53.6% |
| Laetoli A | 161.7 (2.9) | 91.7 (4.7) | 65.0 (0) | 320.0 (12.2) | -24.8 (n=1) | 70.8% | 20.3% | 28.7% | -7.8% |
| Laetoli G1 | 184.8 (6.8)[‖] | 86.8 (4.9)[‖] | 54.4 (4.0)[‖] | 416.0 (38.0)[¶] | 68.0 (23.0)[¶] | 62.7% | 13.1% | 20.9% | 16.3% |
| Laetoli G2/G3[¶] | 208.8 (11.6) | 85.5 (6.3) | 75.9 (5.1) | 433.0 (25.0) | 106.0 (49.0) | 88.8% | 17.5% | 19.7% | 24.5% |
| Laetoli S1[#] | 251.8 | 97.2 | 72.4 | 579.0 | 75.3 | 74.5% | 12.5% | 16.8% | 13.0% |

Values in parentheses represent standard deviations

[*] Data from DeSilva and Gill[19] (Ages 2-7 n=73, Age 8+ n=581)

[†] Data from Hatala et al[20,21]. (n=12 juvenile footprints, n=85 adult footprints)

[‡] Data from Hatala et al[42]. (n=45 footprints)

[§] Data from Thompson et al[28]. (n=18 footprints)

[‖] Data from Hatala et al[42].

[¶] Data from Tuttle[4]

[#] Data from Pelissero[32] (no standard deviations were available)

# Reporting Summary

## Statistics

For all statistical analyses, confirm that the following items are present in the figure legend, table legend, main text, or Methods section.

| n/a | Confirmed | |
|---|---|---|
| ☐ | ☒ | The exact sample size (*n*) for each experimental group/condition, given as a discrete number and unit of measurement |
| ☐ | ☒ | A statement on whether measurements were taken from distinct samples or whether the same sample was measured repeatedly |
| ☐ | ☒ | The statistical test(s) used AND whether they are one- or two-sided<br>*Only common tests should be described solely by name; describe more complex techniques in the Methods section.* |
| ☒ | ☐ | A description of all covariates tested |
| ☒ | ☐ | A description of any assumptions or corrections, such as tests of normality and adjustment for multiple comparisons |
| ☐ | ☒ | A full description of the statistical parameters including central tendency (e.g. means) or other basic estimates (e.g. regression coefficient) AND variation (e.g. standard deviation) or associated estimates of uncertainty (e.g. confidence intervals) |
| ☐ | ☒ | For null hypothesis testing, the test statistic (e.g. *F*, *t*, *r*) with confidence intervals, effect sizes, degrees of freedom and *P* value noted<br>*Give P values as exact values whenever suitable.* |
| ☒ | ☐ | For Bayesian analysis, information on the choice of priors and Markov chain Monte Carlo settings |
| ☒ | ☐ | For hierarchical and complex designs, identification of the appropriate level for tests and full reporting of outcomes |
| ☒ | ☐ | Estimates of effect sizes (e.g. Cohen's *d*, Pearson's *r*), indicating how they were calculated |

*Our web collection on statistics for biologists contains articles on many of the points above.*

## Software and code

Policy information about availability of computer code

| | |
|---|---|
| Data collection | ImageJ (v. 1.49), Teckscan Footmat Research (v. 7.10),  Agisoft Metashape (v. 1.4.4 and v. 1.7.1), Autodesk Meshmixer (v. 3.5.474) |
| Data analysis | R (v. 3.6.1 and v. 4.0.3), Excel (v. 2102), Ultimaker Cura (v. 4.8.0), Cloud Compare (v. 2.11.3), SAGA GIS (v. 7.3.0), ArcGIS (v. 10.6.1), MaxTRAQLite+ (v.2.4.0.3), Geomagic Wrap (v. 2021.0.0), DigTracePro (v. 1.8.1) |

For manuscripts utilizing custom algorithms or software that are central to the research but not yet described in published literature, software must be made available to editors and reviewers. We strongly encourage code deposition in a community repository (e.g. GitHub). See the Nature Portfolio guidelines for submitting code & software for further information.

## Data

Policy information about availability of data

All manuscripts must include a data availability statement. This statement should provide the following information, where applicable:

- Accession codes, unique identifiers, or web links for publicly available datasets
- A description of any restrictions on data availability
- For clinical datasets or third party data, please ensure that the statement adheres to our policy

Previously published data was obtained from ref.4,19-21,28,32,42. The authors declare that all other data supporting the findings of this study are available within the paper [and its supplementary information files], including original source data for figures [1-3], extended data figures [3-6], and extend data table [1]. The photogrammetric reconstruction of the Laetoli A trackway based on three original photographs from the 1977 expedition is available on Morphosource (accession #: in process). The photogrammetric reconstruction of Laetoli A trackway using 57 photographs taken of the re-excavated discovered Laetoli Site A footprints in 2019 is available on Morphosource (accession #: in process).

# Field-specific reporting

Please select the one below that is the best fit for your research. If you are not sure, read the appropriate sections before making your selection.

☐ Life sciences ☐ Behavioural & social sciences ☒ Ecological, evolutionary & environmental sciences

For a reference copy of the document with all sections, see nature.com/documents/nr-reporting-summary-flat.pdf

# Ecological, evolutionary & environmental sciences study design

All studies must disclose on these points even when the disclosure is negative.

| | |
|---|---|
| Study description | This study compared footprint characteristics of black bears (Ursus americanus), chimpanzees (Pan troglodytes), and the fossilized footprints at Engare Sero, Tanzania with previously collected data on the footprints of Pan troglodytes and humans (Homo sapiens) to understand the fossilized footprints from Laetoli, Tanzania. |
| Research sample | The research sample at the Kilham Bear Center consisted of two groups. The first was a set of wild Ursus americanus from the local population in New Hampshire, US that were habituated to allow observation and video recordings. The second was a set of four semi-wild juvenile Ursus americanus (between the ages of 5-8 months). These individuals were chosen to represent the wild black bear population. These juvenile bears were rescued from the wild after being orphaned. These bears are returned to the wild once they reached a sufficient age to survive on their own. These juvenile bears were chosen given their similarity in foot length to the Laetoli Site A trackway. The research sample at Ngamba Island Chimpanzee Sanctuary included the 46 (n=18 male, 28 female; ages 12-36) adult semi-wild Pan troglodytes living at the sanctuary. This population is made up of individuals rescued from the wild due to varying circumstances (e.g., rescued from illegal pet or bushmeat trade). Semi-wild individuals were utilized in this study to avoid some of the changes in behavior and locomotion often present in zoo animals, while still presenting a feasible location for data collection. Data were taken from two published sources examining the bipedal footprint characteristics of two captive subadult male (between the ages of 6.5-7 years old) chimpanzees (Pan troglodytes) housed at Stony Brook University. These data came from Hatala et al., 2016a and Thompson et al., 2018. These 2 sets of chimpanzees were chosen to represent the wild chimpanzee population. The research sample from Engare Sero, Tanzania included a set of 113 footprints (age and sex unknown) belonging to Late Pleistocene Homo sapiens taken from an orthophoto of the site. Footprint characteristics represent fossilized footprints from an unshod/minimally shod population of humans made in similar circumstances to the Laetoli A site. Data were taken on shod human footprint characteristics recorded at the Boston Museum of Science from DeSilva and Gill, 2013. These data included convenience sample of adults (n=581, age 8-80; 366 female, 215 male) and children (n=73, age 2-7; 29 female, 44 male). Cross-stepping data were recorded on 10 adult shod humans (individuals between 19 and 48 years old, with 6 female, 3 male, and 1 non-binary represented) at Chatham University. Data from humans were chosen to represent the broader Homo sapiens population. Data on unshod/minimally shod humans [n=29 adults (15 male, 14 female, aged 18-47) and n=12 juveniles (10 male, 2 female, aged 4-15)] and the Laetoli G footprints (n=5) were taken from Hatala et al., 2013 and Hatala et al., 2016b. Data on the Laetoli S footprints were taken from Pelissero, 2017. The research sample from the Laetoli Site A, included the set of five preserved fossilized bipedal footprints. |
| Sampling strategy | Sample sizes for animal fieldwork were chosen based on the availability of individuals at semi-wild animal sanctuaries. Very little kinematic data exists for wild/semi-wild individuals due to the difficulties of collecting data in the field. The study recruited all individuals available at both the Kilham Bear Center and Ngamba Island Chimpanzee Sanctuary. Sample sizes for fossil footprint trackways (Laetoli A, S, and G) as well as Engare sero were determined based on the preservation of the footprints and availability of data. All viable, undistorted prints were included. For the novel human cross-stepping data, a sample size of 10 subjects producing 10 footprints each was sufficient for a quantitative assessment of how cross-stepping mechanics tended to influence both the perimeter dimensions and the internal topography of the tracks they produced. |
| Data collection | Video data on wild Ursus americanus behavior was recorded by B.H over the course of several years at his ongoing field site in Lyme, NH. These videos were digitized and analyzed by C.J. Data were collected on the juvenile black bears by E.M. and P.K. The juvenile bears were incited to walk independently bipedal across a mud trackway for either a maple syrup or applesauce reward. The mud trackway was then removed from the enclosure and the defined footprint characteristics were recorded. Data were collected on adult Pan troglodytes individuals at the the Ngamba Island Chimpanzee Sanctuary by E.M. Individuals walked across a plantar pressure mat placed in the walk way connecting their overnight enclosure and the open forest habitat. Individuals passed over the mat twice a day as the entered and exited the forest. Data on the Engare Sero footprint trackways were collected by C.L.-P., B.Z., and E.K. An orthophoto was generated of the site by by C.L.-P. and B.Z. Footprint characteristics were then measured on this orthophoto by E.K. using ImageJ. Data were collected on the Laetoli Site A prints by A.S.D., K.F., L.D.F., J.G., E.G., D.K., B.M., A.P., S.R., R.T., C.M.M., and J.M.D. who relocated, re-excavated, and measured the footprint characteristics of the original Site A trackway. Data were collected by K.H. and E.M.W on habitually shod humans walking with both their normal gait and a cross-stepping gait. The defined footprint characteristics were recorded and analyzed. |
| Timing and spatial scale | Data on the semi-wild chimpanzee at Ngamba Island Chimpanzee Sanctuary were recorded between December 9-16, 2018. Length of sampling was determined by funding and availability of the sanctuary. Data were collected twice a day; once in the morning (between 6:45-8:00 am) when the chimpanzees were headed to the forest for their first feeding, and once at 6:00 pm when the chimpanzees were headed into the overnight enclosure to sleep and receive their last feeding. Data on the juvenile bears at the Kilham Bear Center were recorded over the course of four visits in 2017 (May 11, Aug. 11, Aug. 14, Aug.31st). Length of sampling was determined by availability of researcher and sanctuary. Data were recorded in the morning/early afternoon for each visit so that data collection could occur prior to their feeding. This was to encourage participation as food was used as a reward. The Engare Sero orthograph was constructed using photographs taken between June 2-17, 2010. Multiple days were necessary to construct this composite image due to its size and variation in weather (i.e., some days were too windy to keep the tent over the camera. Data analysis on the Engare Sero footprints were collected and analyzed periodically between September 2019 and April 2020 as the work was being completed as part of E.K.'s graduate research. Data on the Laetoli Site A trackway were collected between June 19-25th, |

2019. Field season length was determined by funding and availability of researchers. Cross-stepping data were recorded between June 12-22, 2021 as that was sufficient to complete collection across the 10 participants.

| | |
|---|---|
| Data exclusions | Footprint data were only excluded from the study if there was insufficient preservation of the track to measure the characteristics of interest defined in the methods. (E.g., a footprint might be excluded if it does not preserve the heel impression and thus cannot have the foot length measured). |
| Reproducibility | Footprint characteristics were defined prior to the start of the study. All novel footprints were preserved (including plaster casts of bear prints, 3D surface models of Laetoli Site A, the original in situ Site A prints, and plantar pressure impressions of Pan troglodytes) so measurements can be repeated by other researches. 3D surface scans of both the 1977 excavation and our 2019 re-excavation are publicly available on Morphosource for other researchers to access and validate our measurements. Reproducibility for individual footprints across our comparative sample (humans, bears, and chimpanzees) was established by collecting and comparing multiple footprints of the same individuals. |
| Randomization | Randomization was not relevant to our study as we were interested in measuring footprint characteristics from whole sample populations, as opposed to comparisons within those populations. |
| Blinding | Blinding was not relevant to the data collected on the non-human comparative species (e.g., bears and chimpanzees) nor to the data collection on fossilized footprints. The human participants were unaware of the site A tracks at Laetoli and therefore had no knowledge of how the data obtained from their footprints would be used in this study. |

Did the study involve field work?  ☒ Yes  ☐ No

## Field work, collection and transport

| | |
|---|---|
| Field conditions | Fieldwork was completed at two semi-wild animal sanctuaries, one in Entebbe, Uganda and one in New Hampshire, US. Additional fieldwork was conducted at Laetoli, Tanzania. For fieldwork completed at both animal sanctuaries, environmental conditions were unlikely to effect the outcome of the experiment. For both locations, data collection occurred during clear weather at temperatures that were within seasonal norms. For the fieldwork at Laetoli, the weather was clear and within seasonal norms and thus unlikely to have impacted data collection. |
| Location | Fieldwork was conducted at three locations: 1) the fossil site of Laetoli, Tanzania (S 03.13.185' E 035.11.976' taken on-site with Garmin GPS) during June 2019; 2) Ngamba Island Chimpanzee Sanctuary located on chimpanzee island (-0.10409381597120933, 32.652780666056096 from Google Map) near in Entebbe, Uganda in December 2018; and 3) the Kilham Bear Center (43.770876615493975, -72.09782685342287 from Google Map) in Lyme, NH in 2017. |
| Access & import/export | Fieldwork was conducted at several locations. Research permits were granted by the Tanzanian Commission for Science and Technology (permit 2019-370-NA-2019-2016) to access the site of Laetoli in Tanzania. Permission to study the chimpanzees in Uganda was provided by the Uganda Wildlife Authority (UWA/COD/96/05) and the Ugandan National Council for Science and Technology (NS65ES). No materials were imported/exported. Permission to study the black bears at Kilham Bear Center was provided by the sanctuary director, B. Kilham. No animal materials were exported/imported. All animal protocols were approved by the Dartmouth College Institutional Animal Care and Use Committee. |
| Disturbance | The study exposed the surface of the fossilized footprints at Site A in Laetoli, Tanzania by disturbing the top soil covering this trackway. This disturbance was minimized by the careful reburial of the trackway to prevent potential damage to the footprints. The study caused a slight disturbance to the daily routine of the chimpanzees at Ngamba Island Chimpanzee sanctuary and the black bears at the Kilham Bear Center. This disturbance was minimized through study design input by the Sanctuary and Center staff as well as efforts to habituate the individuals to the presence of the research and plantar pressure mat. |

# Reporting for specific materials, systems and methods

We require information from authors about some types of materials, experimental systems and methods used in many studies. Here, indicate whether each material, system or method listed is relevant to your study. If you are not sure if a list item applies to your research, read the appropriate section before selecting a response.

## Materials & experimental systems

| n/a | Involved in the study |
|---|---|
| ☒ | Antibodies |
| ☒ | Eukaryotic cell lines |
| ☐ | ☒ Palaeontology and archaeology |
| ☐ | ☒ Animals and other organisms |
| ☐ | ☒ Human research participants |
| ☒ | Clinical data |
| ☒ | Dual use research of concern |

## Methods

| n/a | Involved in the study |
|---|---|
| ☒ | ChIP-seq |
| ☒ | Flow cytometry |
| ☒ | MRI-based neuroimaging |

## Palaeontology and Archaeology

| | |
|---|---|
| Specimen provenance | We received approval from the Tanzania Commission for Science and Technology (permit 2019-370-NA-2019-2016) to conduct |

| Specimen provenance | research at site of Laetoli. |
|---|---|
| Specimen deposition | The fossil footprints remain in situ at site A in locality 7 at Laetoli, Tanzania. The Site A footprint excavation has been reburied to protect the footprints and the site marked. |
| Dating methods | no new dates are provided |

☐ Tick this box to confirm that the raw and calibrated dates are available in the paper or in Supplementary Information.

| Ethics oversight | Ethical oversight of the research project was provided by the Tanzania Commission for Science and Technology. |
|---|---|

Note that full information on the approval of the study protocol must also be provided in the manuscript.

## Animals and other organisms

Policy information about studies involving animals; ARRIVE guidelines recommended for reporting animal research

| Laboratory animals | The study did not involve laboratory animals |
|---|---|
| Wild animals | Study examined four semi-wild juvenile black bears (Ursus americanus) housed at the Kilham Bear Center in Lyme, NH awaiting reintroduction into the wild. These individuals (3 male, 1 female) were released back into the wild (Vermont and New Hampshire) after study completion through cooperation between the Kilham Bear Center and New Hampshire Fish and Game Department. The study followed the bears between the age of 5 to 8 months old. Additionally, this study examined just over 50 hours of video data recording Ursus americanus behavior on wild black bears in New Hampshire, US. These included both adult and juvenile individuals (precise age and gender unknown). Study also examined 46 adult (n=18 male, 28 female; ages 12-36) semi-wild chimpanzee (Pan troglodytes) individuals housed at the Ngamba Island Chimpanzee Sanctuary in Entebbe, Uganda. These individuals remain at the Ngamba Island Chimpanzee Sanctuary. No individuals were captured in the field or transported for the study. |
| Field-collected samples | The study did not involve samples collected in the field. |
| Ethics oversight | Ethical approval was provided by the Dartmouth College Institutional Animal Care and Use Committee. |

Note that full information on the approval of the study protocol must also be provided in the manuscript.

## Human research participants

Policy information about studies involving human research participants

| Population characteristics | Data were collected from 10 healthy adults (including 6 female, 3 male, and 1 non-binary between 19 and 52 years old) without foot or other lower limb maladies that might affect their mobility. Neither age nor gender was expected to affect the hypotheses being tested (how footprint morphology changes due to cross-stepping). |
|---|---|
| Recruitment | Participants were recruited via email and by word-of-mouth (snowball sampling). We did not identify the potential for any form of recruitment bias that may impact results. |
| Ethics oversight | Ethical approval was provided by the Chatham University Institutional Review Board |

Note that full information on the approval of the study protocol must also be provided in the manuscript.

