## [Peer Review File · Nature]

Manuscript Title: Footprint Evidence of Early Hominin Locomotor Diversity at Laetoli Tanzania

Reviewer Comments & Author Rebuttals

Reviewer Reports on the Initial Version:

Referee #1 (Remarks to the Author):

This well written and enjoyable paper makes two arguments: (1) the five putative hominin footprints from the A trackway at Laetoli were not made by bears, and (2) that the footprints (mostly the A3 footprint) have a wide forefoot relative to total foot length and more similar to those of chimpanzees than the hominin that made the famous G trackway (presumably *A. afarensis*). They suggest that two species of hominins are represented at Laetoli, with the A trackway maker potentially being something like whatever species belong to the Burtele foot.

In terms of the ursine hypothesis, I think we can say the bear hypothesis can exit stage left (sorry) because this paper makes a thorough, solid case based on numerous lines of evidence including the unlikelihood of bears taking 5 consecutive bipedal steps, the absence of ursid fossils from Laetoli, the absence of claw prints, the relative size of the digits, and the fact that the maker of this track was apparently cross-stepping.

The second hypothesis is certainly reasonable but less well securely tested. The strongest evidence for A3 being made by a different species than the maker of the G trackway is the width of the forefoot relative to the hindfoot for the A3 print. However, with just one print, there is no confidence interval for this assessment. Less strong is the evidence of hallucal divergence and the difference between this footprint and those made by humans today and in the past. Based on these data, I don't disagree with the authors, but I worry about how secure the conclusions are given how little information there is to analyze. Another worrisome issue is how cross-stepping affects footprints. Since none of the reference samples collected were of cross-stepping individuals, how do we know cross-stepping didn't influence the shape of the footprints, especially for a species whose locomotor anatomy was different from a modern human? Also, why is there no analysis of the ratio of toe to heel depth, as done so effectively by Raichlen et al (2010, Plos One) for the Laetoli A footprints?

So, on the whole, this is an interesting paper and it certainly merits being published in a good journal, but I don't think the refutation of the ursine hypothesis makes it strong enough for Nature. The evidence to support the hypothesis that a species other than *A. afarensis* made the A trackway rests almost entirely on a single footstep from a non-human hominin taking short cross-steps but we don't know what an *A. afarensis*' footprints look like when it is taking short cross-steps in part because the study doesn't examine what such footprints would look like in either chimpanzees or humans.

Referee #2 (Remarks to the Author):

This manuscript presents research on a series of footprints from Laetoli (Track A). The authors bring new comparative data to the question of which animal made the track, and the related topics of gait and foot anatomy. Another novel contribution is that they examine their results in light of recent evidence of hominin taxonomic diversity and/or locomotor diversity that was not known when the Laetoli footprints were initially discovered.

This is an exciting and thought-provoking topic! It will be of great interest to the

paleoanthropological community. The notoriety of the Laetoli footprints will likely make the paper of interest to the more general audience of Nature. The article is very well written. I would recommend publication, pending major revisions that include adding new comparative data.

- The authors use Russ Tuttle's hypothesis framework (Lines 76-87) to summarize previous opinions on Track A and to structure their manuscript, which I found effective. The primary strength of this manuscript is in refuting hypothesis #2, that the footprints were made by a juvenile ursid.

- o Figure 2B is simple but very convincing.

- o The authors also make a new and effective argument in lines 196-208. However, I had to read this section a couple times before I got it. Consider editing the beginning as: "Facultatively bipedal primates like chimpanzees produce..."

- o Another point that would benefit from clarification is why the ursid 4th & 5th rays are compared to primate 1st & 2nd rays. Are homologous rays numbered differently? Or is that about siding the ursid tracks?

- A good chunk of the manuscript deals with hypothesis #3, that the footprints were made by a different kind of hominin, with a different bipedal gait, than those at Site G. To address this hypothesis, the authors compare Laetoli Track A to Tracks G and S, and to human and chimp footprints and pressure mat data. Within Tuttle's hypothesis #3, I see two possibilities: Track A represents an *A. afarensis* individual walking in an atypical manner or it is another hominin species (i.e. one with different foot anatomy and locomotor behavior). The authors support the latter interpretation but do not present data to refute the former. The primary weakness of this manuscript is that there are no comparative observations on what the footprints of a "cross-stepping" human look like. This manuscript would benefit greatly from the addition of such data.

- o Footprints are the result of the three-way interaction of anatomy, locomotor forces, and substrate properties. The cross-stepping gait could change the way that weight is transferred across the mid- and forefoot (from ML rocking or the addition of more rotational component to the swing phase side of the body) that might widen a cross-stepping footprint, when compared with footprints generated through normal, forward walking. In order to argue that this A3 footprint shape indicates different foot anatomy (as the authors do), it is necessary to somehow control for the effect of cross stepping. Specifically, it would be good to provide evidence about:

- ♣ if/how the Mahalanobis distance (using the topographic data, as in Fig. 3C) is increased when comparing normally walking and cross-stepping human footprints

- ♣ if/how the hallucial divergence ratio differs between normally walking and cross-stepping human footprints

- ♣ These data shouldn't be extracted from a pressure mat—the mud aspect would seem to be important to the slippage and sliding factor

- o I didn't understand what was meant by "cross-stepping" until I tried walking like this (only then did Leakey's line 67-69 description make sense for me). It is better explained near the end of the manuscript (that the foot from each side crosses the midline before touchdown) and authors might consider moving that earlier.

- ♣ Lines 191-195: How would a footprint and track be affected if someone were walking forward, then looked behind them (in both directions) while continuing forward?

- Another issue that relates to the hypothesis #3 interpretation is the difference between the shape of a single footprint and properties of a track (something measured among prints). These are different concepts to me, but I had the impression that "footprints" is used to cover both in this manuscript. This is relevant because the authors argue that Laetoli A differs from Laetoli G and S when track properties are considered (Fig 2D, Supp Figs 3D, 3A and 3B) but it is much less clear that the difference is present when the shape of a single footprint is considered (Supp Fig 3C, Supp Table 1 and actually even Supp Fig 4 gives that impression). For these reasons, I thought that the data presented came a bit short of demonstrating the authors' preferred interpretation—

that this is the footprint of a grasping toe hominin.

o As a related point, The authors talk about the Burtele foot in relationship to footprint A3. It would be good to add something about how ray 1 length of the fossil and the footprint compare.

o It would be valuable for the authors to add a clear statement about whether they consider Track A to represent the "normal" gait of this other grasping toe hominin species or whether they consider this gait to be atypical (setting foot anatomy aside).

Smaller points:

Figure 1 – it would be nice to have a sense of the spatial relationship between Track A and Tracks G/S. Is it possible to include that in insets C or D?

Figure 1 – it would be useful to label the five footprints in the middle panels

Line 118: check the figure call out (supposed to be Supp Fig 1 or 2?)

Line 124: check the figure call out (supposed to be Supp Fig 2?)

Figure 2 Legend line 303: where bears walked four or more steps... ; line 312 remove "with" possibly rephrase as 'Ratio of stride width to step length across different...'

Figure 2B are these both left footprints?

Supplemental Table 1 – is it possible to include measures of variation?

Supplemental Figure 3 – Are these comparisons all based on just one observation from the A3 print? What about the 5th print, which belongs to the same track but looks quite narrow?

In all figure and table legends, it would be useful to indicate whether the data come from chimps walking quadrupedally vs bipedally, or to indicate if these data are mixed. This info isn't readily apparent from the supplemental text.

It would be good to include a sentence about how actual footprint measurements compare to those taken from pressure mat data.

Lines 226-230: it would be useful to add a comparison of a chimp/human/Laetoli footprint to figure 3 (or Supp Figure 4) and one that ideally showing how a mid-tarsal break looks in footprint form.

Referee #3 (Remarks to the Author):

The manuscript titled "Footprint evidence for early hominin locomotor diversity – Laetoli, Tanzania" re-examines the Site A tracks which were originally discovered in 1976. The authors relocated and re-excavated the footprints in 2019, and conducted dimension-based assessments to determine if the tracks were hominin or if they could have been made by an ursid walking bipedally. In conclusion, they argue that the prints were likely made by a hominin. I commend the authors for their diligence and efforts in the relocation of the footprints first discovered and then reburied almost 45 years ago. I also applaud their data collection because it is quite an admirable sample of extant data which has been included in the study, alongside some very beautiful diagrams/figures.

Originality and significance: the manuscript is not 'traditionally' original because the prints were discussed in the 1970s, but the authors' relocation of the fossils, excavation and new analyses are of significance to the field. In palaeoanthropology, there are such few footprint data out there. If these prints are truly hominin then it would (1) boost our current dataset for hominin trackways, and (2) most importantly: could argue the case for two bipedal hominins during the Pliocene in Eastern Africa. The latter of which is very exciting.

I have both minor and major comments of the study which I believe should be addressed prior to publication of the manuscript. The minor comments regard clarification of a few sentences, typographical errors, additional references to cite, and some necessary tweaks to a few figures for better readability. My major comments mostly regard the following: (1) the assumption that no erosion had occurred between 1976 and 2019, alongside a recommendation for an additional [quick] extra analysis to somewhat explore erosion of the bed/prints, (2) upon my examination of

the Site A prints in the depth maps, I am unconvinced of the cross-cutting story and instead believe that these two prints may not belong to the same trackway – I recommend the authors conduct an MNI of the full trackway to determine the likelihood that all of these prints belong to the same individual and were made at the same time, (3) the ursid measurements must be included in the comparative analyses + statistics if the authors wish to establish if the prints are ursid or not, (4) I appreciate that the footprints are not in the best preservation condition, but as many footprints as possible must be included in the comparative analyses, there cannot be a sample size of one in the comparative analyses, and finally (5) the distance between the first and second digits is quite large and, therefore, I would urge the authors to calculate the hallucal angle for each of the Site A prints to determine if the angle falls within human/hominin ranges. This additional analysis would provide extra support to their conclusions.

Minor comments:

Lines 143-146: I understand what the authors are trying to say, but I think that the wording of this sentence could be made clearer.

On line 146 – please clarify the statement “no fossil ursids...”. It seems like a pre-emptive conclusion that the prints are not ursid, but hominin. Please clarify that you (assumedly) mean no skeletal material.

Line 147: “25,000 fossils and 85 mammalian species” – do the authors mean that there are 25000 fossils attributed to 85 mammalian species? Please clarify this statement.

Lines 156-157: please clarify if you mean the Site A or Site G prints.

Line 158: “semi-wild juvenile black bears”, please specify species name. Please also provide ball-park figures for the statement that their foot sizes were similar in length to the track sizes. I appreciate that this can be found in the supp info, but it should also be included here using minimum amount of words.

Lines 172-173 – Do the authors mean as in humans when taking short steps, or ursids? Please clarify.

Throughout manuscript: earlier in the manuscript, the authors state via footnote that they will use the terminology ‘hominin’ but interchangeably use the term ‘hominid’ and ‘hominin’. Because Nature has a wide readership (many of whom will presumably not be palaeoanthropology specialists), I urge the authors to either (1) change terminology to be consistent or (2) define why you are using the term ‘hominid’ versus ‘hominin’. The current footnote is not explanatory enough to a lay-reader.

Lines 183: I’m personally not too sure what the phrase “scalloped impression” means as I have a few ideas of what this could look like when imagining it. Can a different terminology/phrase be used?

Line 195: “we did not observe cross-stepping in our bear sample” – does this refer to quadrupedal steps or bipedal steps or both?

Paragraph beginning line 196: Are the authors referring to the pathological gait deformity called valgus knee? Or is it rather for the description of limb posture in hominins? If the former, then please provide reference(s) for valgus knee pathologies. If the latter, then please clarify in text that it is not a pathological description/interpretation. Perhaps a short explanation describing that the term valgus knee is related to the bicondylar angle/posture of the lower limb.

Line 225: Can the figure/photograph of the footprint A3 in the supp info be cited at this point?

Line 235: Please add "Dikika', Afar, Ethiopia".

Line 251: Please add the word 'and' between "distinct, presumably...".

Should the text after line 256 come under a new sub-heading?

Line 291: Should Google Maps either have a citation or a (C) (copyright) symbol?

Figure 2: the % times are reported in text, there is no need for panel A. Please fix x-labels in panel D so that they are all aligned equally, i.e., "modern shod humans". If it is difficult to align the text due to lengths of the labels, then consider shortening label names and clarifying in the figure caption. [this also applies to SI Figure 3 boxplots]. Additionally, please re-iterate in the figure/figure caption the sample size for each group. Panel B – the right ursid print seems to be cropped too closely to the border of the print and, as such, some data could be missing. Please see Falkingham's 2014 paper on objective track outlines. Panel C – please make this colour-blind friendly. Whilst the orange and the blue are fine, the blue on the black will not be readable by some (i.e., the median line in the boxplot).

Line 371: please reword this sentence to state that it is not the same chimpanzee, but the same species (I assume?).

Line 371: MaxTraQ and ImageJ need to be cited.

Line 414: add space between "581individuals"

Line 449 + 477: please specify if specific R packages were used? I assume Figure 3 was ggplot?

For clarity in the supp info, a table should be created listing the samples/specimens used. This would benefit the reader for quicker recap when examining the graphs.

Line 452: Because the analyses are a pivotal part of the paper, they should be briefly re-discussed here rather than referring the reader to another paper for the methods.

Line 473: for all Laetoli tracks? Or just the Site G or Site A? Please specify.

Line 485: please specify why the photographs were taken in the morning? Less glare from the sun? More shade?

In SI Figure 2: I like the diagrams! However, the blue shapes for digits 3, 4 and 5 haven't fully "closed". Is it possible to remove the rogue blue lines extending from the shapes? Additionally, I think the arrows highlighting the midtarsal break put a bit too much emphasis on the ridge, whilst the manuscript contrastingly alludes to the ridge very briefly. The arrows should be replaced with a shape, a line or just one arrow to 'soften' the figure's argument.

I very much like the supplementary pdf of the 3D model. However, can the underside of the model be given more colour? Currently when I rotate the model using Adobe, I am unable to clearly see the underside due to the lighting. Will the model also be made available upon publication to download?

The Miocene footprints from Crete also investigated if the prints were hominin or ursid. This paper should be referenced here (Gierlinski et al. 2017; Crompton 2017).

Major comments:

As I am understanding it, based upon qualitative assessment of the tracks, an assumption was

made that little to no erosion occurred? Whilst I agree that the data does not exist to quantitatively assess erosion of the prints, I do not quite agree with the statement that no erosion occurred. This statement should be softened and acknowledged that there is scope for some erosional processes to have occurred as has been studied elsewhere (citing Zimmer et al. 2018; Wiseman et al. 2018 whom both quantitatively examined erosional processes in footprints). In the supplementary information the authors discuss how their measurements vary versus those collected in the 70s by Tuttle. All measurements now are smaller than Tuttle's. This appears to me to be the cause of erosion, although a stronger rebuttal and additional evidence from the authors could be persuasive enough to challenge my concern – for example, a quick mesh to mesh distance analysis either in R or CloudCompare could be computed using the 3D models to visually examine the shape differences between 1976 and 2019 (with an explanation for the regions that have had additional debris removed).

Figure 1: Panels D and E are hard to read with the white text on the black background. The colours should be inverted for ease of readability. Also, it is not entirely clear what the orange rectangle in panel D is showing – I presume it is highlighting where panel E is located? Perhaps panel E could become a 'pop-out' box (i.e., in which lines connect the edges of the rectangle to panel E) from panel D for clarity? Additionally, the text in panel C needs to be bigger. Is the example of cross-stepping shown in the middle panels? If so, those cross-steps are quite extreme and rather look like they could belong to different individuals. How sure are the authors that this is cross-stepping? Do the footprints show evidence of slippage which could correspond to cross-stepping? Assuming that the base of support lies medially to the right footprint, that cross step is what I would expect the maximum bounds in an anatomically modern human; I'm sceptical that is a cross-step in a hominin which may not have adducted hips + valgus knee, particularly without additional trace evidence of slippage. This argument requires further evidence to back up the claim. Otherwise, the authors should soften their argument and state that it could either be cross-cutting evidence or could potentially be footprints belonging to two individuals not within the same trackway. The right footprint also looks a bit problematic in which the heel looks a bit too pointed and toe line too straight – which leads me back to my original comment: how sure are the authors of minimal erosion of the prints? But also – are the authors certain that the shapes of the footprints are hominin and not ursid? Can additional analyses be computed to convince the reader, such as shape analyses? I would also argue for the presence of an unidentified footprint in that middle panel – there appears to be a print crossing almost diagonally across the top. The shape of the print on the colour-rendered map is very similar to human prints from Namibia and Formby Point. Has the diagonal impression been investigated? If so, then I trust the opinion of the authors whom have the 3D data in front of them, but (if so) I would also urge the authors to add an additional section to the supp info on other hollows in the area. For example, could the other impressions on the right side of the panels also be potential prints? And what methods did you use to exclude them from the conclusion that a single trackway was present? My opinion based upon these images is that there are more than one individual present and this needs to be investigated. What is the MNI based upon measurements/shape? Finally, I would recommend to explicitly highlight the tracks on the middle panels for clarity. Towards the top of the panel they become a bit harder to discern and I think the reader would benefit from them being highlighted/circled/numbered/etc.

Figure 3: because this paper is investigating if the prints were ursid or not, ursid measurements need to be included in this figure + analysis. In panel A – has the site G photograph been reproduced from another publication? Or used with permission from someone? Please state in figure caption if so. Additionally, please rearrange the labels in panel C. It is currently hard to read "Laetoli G" for example, due to the label's close proximity to the dashed lines.

Apologies, but I have to disagree with the authors that that black bear and Site A prints are similar in size. 145mm versus 161mm is quite a difference; in human children this is a difference of 5 shoe sizes (i.e., assuming half sizes are included – but could be argued for just two shoe sizes difference –so a size 7 versus a size 9; measurements calculated using startriteshoes.com/size-calculator). I do not argue that this discrepancy affects their analyses, but the wording should be

changed. Additionally, because the analyses rely on the fact that feet are so similar in size, I would recommend that the data should be normalised prior to statistical analyses (sufficient to use ratios).

Line 456: A sample size of one is insufficient to be conducting analyses, and I am surprised by the authors' decision to exclude most of a readily available sample. Despite the other tracks being of 'poorer quality', as much data as possible from Site A must be included. From visual inspection of the figures, I would argue that at least three of the prints are usable. Otherwise, the morphology from this one print could be driven by erosion, poor definition, or just simply an 'anomaly' of a print. McClymont and Bennett (2021) (see also McClymont's PhD thesis; U of Liverpool 2016) established that a sample size of >250 is necessary to make inferences from footprints. But please also see Belvedere et al. (in press; *Palaeontology – Questions of sample in ichnology*; please contact authors if not yet published by the time revisions are being done for paper sharing) in which a minimum sample size of seven was required to make inferences from footprints based upon change-point modelling. Whilst I completely understand that this is an impossible sample size for the current study due to only five fossil footprints being present, I must urge the authors to include more than one footprint in their comparative analysis here. This is particularly pertinent due to the analysis regarding depth measurements – some footprints are light, some footprints are deep. With a sample size of just one, then the spectrum is lost. More-so, if there is evidence of cross-stepping then I would assume that the plantar pressure on the cross-stepping foot would be greater, thus skewing results. If there is any slippage in the footprint then this may further affect the results. To give a broad overview into the Site A footprints (for the first time) then more footprints must be included in the comparative analyses.

Regarding the depths of the prints, please state how the depths were measured. Was this the absolute depth from the surface? How was this defined? Was a plane created?

Line 466: in the creation of the mean human footprint, please cite Bennett et al. 2016 and Belvedere et al. 2018 in which both of those papers created an averaged track. Please also clarify what is meant by stating that it is not an actual footprint? Is it composed of different measurements? Shape differences could be quantified using DigTrace Pro/Academic which is freely available software and could provide an averaged 3D track as a .obj file. That 3D model could then be compared to the Site A print via the same method of track registration.

Regarding the photogrammetry: Please state what different software was attempted to reconstruct the prints – I assume more than one? Although I do applaud the authors for managing to create a 3D model in Agisoft. Peter Falkingham has a great online (free) blog of different photogrammetry techniques and software that can be used (<https://peterfalkingham.com>). Some software computes un-distortion of images which may produce more precise models. Additionally, please compare the older 3D models with the re-excavated versions (perhaps even using a simple mesh-to-mesh distance analysis) to examine how similar/dissimilar the prints are. This may assist in identifying erosion.

Line 602: 13mm between the first and second digits is quite large! The paper does not discuss hallucal ad/abduction, but I urge the authors to incorporate this into their manuscript. I would also be interested to know the ranges of hallucal angles in the Site A prints and how they compare with the other Laetoli prints but also with humans. I believe that is a stronger story than inferences upon knee valgus which cannot be supported without biomechanical assessments.

SI Figure 3: in panel D please either change the size, colour or shape of the icons representing the Laetoli samples. It was quite difficult to easily locate Site A on the graph (and in fact I only found it after reading the figure caption), and the other shapes were equally as hard to see too. I would also be careful about making too many strong assumptions based upon this graph. There is overlap amongst all groups on that graph, and the sample sizes for the Laetoli prints is quite small. Additionally, the negative value for the cross-stepping is problematic. I'm not convinced of

the cross-stepping story and would like to see more evidence to support it. If cross-stepping can be supported with more evidence, then I wouldn't see the issue in using absolute values in this graph so long as it was clearly stated in the figure caption that you have done so. Otherwise, this graph does not contribute much to the story of the paper in which the reader wants to see where Site A falls with respect to the other taxa.

SI Figure 4 and the accompanying analyses should include ursid data.

Overall, I enjoyed reading the paper and especially enjoyed the detailed story about the re-excavation of the prints which is often lacking in other papers. I do believe that this paper is publishable after revisions.

DARTMOUTH

Author Rebuttals to Initial Comments:

Referees' comments:

Referee #1 (Remarks to the Author):

This well written and enjoyable paper makes two arguments: (1) the five putative hominin footprints from the A trackway at Laetoli were not made by bears, and (2) that the footprints (mostly the A3 footprint) have a wide forefoot relative to total foot length and more similar to those of chimpanzees than the hominin that made the famous G trackway (presumably *A. afarensis*). They suggest that two species of hominins are represented at Laetoli, with the A trackway maker potentially being something like whatever species belong to the Burtele foot.

We do not yet have the evidence to assess whether there is any close relationship between the Laetoli Site A footprints and the Burtele foot. Our point in bringing up that foot fossil was simply to note that our interpretation of taxonomic and locomotor hominin diversity is corroborated by the existing fossil record.

In terms of the ursine hypothesis, I think we can say the bear hypothesis can exit stage left (sorry) because this paper makes a thorough, solid case based on numerous lines of evidence including the unlikelihood of bears taking 5 consecutive bipedal steps, the absence of ursid fossils from Laetoli, the absence of claw prints, the relative size of the digits, and the fact that the maker of this track was apparently cross-stepping.

Thank you. We agree that the ursid hypothesis is no longer tenable.

The second hypothesis is certainly reasonable but less well securely tested. The strongest evidence for A3 being made by a different species than the maker of the G trackway is the width of the forefoot relative to the hindfoot for the A3 print. However, with just one print, there is no confidence interval for this assessment. Less strong is the evidence of hallucal divergence and the difference between this footprint and those made by humans today and in the past. Based on these data, I don't disagree with the authors, but I worry about how secure the conclusions are given how little information there is to analyze.

Because of this comment and those made by the other reviewers, we have included print A2 in our multivariate analysis of footprint shape. The new results are even more robust: the A footprints are as different from the G or S trackways as a chimpanzee foot is from a modern human foot (see new Figure 3C).

Another worrisome issue is how cross-stepping affects footprints. Since none of the reference samples collected were of cross-stepping individuals, how do we know cross-stepping didn't influence the shape of the footprints, especially for a species whose locomotor anatomy was different from a modern human?

DARTMOUTH

Thank you for making this suggestion. We have now conducted an experiment testing whether footprints made by individuals (N=10) cross-stepping are fundamentally different from those made by individuals walking with their preferred gait. We find that while there are subtle changes to the footprint that can be attributed to cross-stepping, the differences we find in the bipedal footprints from Site A compared with the G or S prints cannot be explained by cross-stepping alone.

Also, why is there no analysis of the ratio of toe to heel depth, as done so effectively by Raichlen et al (2010, Plos One) for the Laetoli A footprints?

Thank you for this suggestion. We have now added this analysis based on the updated proportional toe depth metric proposed in Raichlen and Gordon's (2017) paper to the main text and supplementary methods of the paper.

So, on the whole, this is an interesting paper and it certainly merits being published in a good journal, but I don't think the refutation of the ursine hypothesis makes it strong enough for Nature. The evidence to support the hypothesis that a species other than *A. afarensis* made the A trackway rests almost entirely on a single footstep from a non-human hominin taking short cross-steps but we don't know what an *A. afarensis*' footprints look like when it is taking short cross-steps in part because the study doesn't examine what such footprints would look like in either chimpanzees or humans.

We hope that the modifications we have made to the paper change the opinion of this reviewer.

Referee #2 (Remarks to the Author):

This manuscript presents research on a series of footprints from Laetoli (Track A). The authors bring new comparative data to the question of which animal made the track, and the related topics of gait and foot anatomy. Another novel contribution is that they examine their results in light of recent evidence of hominin taxonomic diversity and/or locomotor diversity that was not known when the Laetoli footprints were initially discovered.

This is an exciting and thought-provoking topic! It will be of great interest to the paleoanthropological community. The notoriety of the Laetoli footprints will likely make the paper of interest to the more general audience of Nature. The article is very well written. I would recommend publication, pending major revisions that include adding new comparative data.

- The authors use Russ Tuttle's hypothesis framework (Lines 76-87) to summarize previous opinions on Track A and to structure their manuscript, which I found effective. The primary strength of this manuscript is in refuting hypothesis #2, that the footprints were made by a juvenile ursid.
 - o Figure 2B is simple but very convincing.
 - o The authors also make a new and effective argument in lines 196-208. However, I had to read this

DARTMOUTH

section a couple times before I got it. Consider editing the beginning as: “Facultatively bipedal primates like chimpanzees produce...”

We have made this recommended change.

o Another point that would benefit from clarification is why the ursid 4th & 5th rays are compared to primate 1st & 2nd rays. Are homologous rays numbered differently? Or is that about siding the ursid tracks?

Thank you for this question/comment. In the third paragraph of the introduction, we write: “Furthermore, the fifth digit is typically the longest in ursids, solving the “cross-stepping” problem of the Site A footprints.” In other words, the Site A individual was either a cross-stepping primate or an ursid not cross-stepping. We have added the following text: “The A3 footprint is thus either a left print of a primate or the right print of an ursid.” As the reviewer suggests, then, it is about siding of the tracks.

- A good chunk of the manuscript deals with hypothesis #3, that the footprints were made by a different kind of hominin, with a different bipedal gait, than those at Site G. To address this hypothesis, the authors compare Laetoli Track A to Tracks G and S, and to human and chimp footprints and pressure mat data. Within Tuttle’s hypothesis #3, I see two possibilities: Track A represents an *A. afarensis* individual walking in an atypical manner or it is another hominin species (i.e., one with different foot anatomy and locomotor behavior). The authors support the latter interpretation but do not present data to refute the former. The primary weakness of this manuscript is that there are no comparative observations on what the footprints of a “cross-stepping” human look like. This manuscript would benefit greatly from the addition of such data.

o Footprints are the result of the three-way interaction of anatomy, locomotor forces, and substrate properties. The cross-stepping gait could change the way that weight is transferred across the mid- and forefoot (from ML rocking or the addition of more rotational component to the swing phase side of the body) that might widen a cross-stepping footprint, when compared with footprints generated through normal, forward walking. In order to argue that this A3 footprint shape indicates different foot anatomy (as the authors do), it is necessary to somehow control for the effect of cross stepping. Specifically, it would be good to provide evidence about:

- ♣ if/how the Mahalanobis distance (using the topographic data, as in Fig. 3C) is increased when comparing normally walking and cross-stepping human footprints
- ♣ if/how the hallucial divergence ratio differs between normally walking and cross-stepping human footprints
- ♣ These data shouldn’t be extracted from a pressure mat—the mud aspect would seem to be important to the slippage and sliding factor

We are grateful for this suggestion and have responded by adding data comparing footprints made by individuals (N=10) walking with a preferred gait to footprints made during cross-stepping. While subtle differences in print morphology occur, these do not match in magnitude or direction the differences between the Laetoli Site G/S prints and the Site A prints. Therefore, we can reasonably

DARTMOUTH

refute the hypothesis that the differences are a result of the same species of hominin walking in a biomechanically different manner (i.e., cross-stepping).

o I didn't understand what was meant by "cross-stepping" until I tried walking like this (only then did Leakey's line 67-69 description make sense for me). It is better explained near the end of the manuscript (that the foot from each side crosses the midline before touchdown) and authors might consider moving that earlier.

We have added a parenthetical definition of cross-stepping earlier in the paper as suggested.

♣ Lines 191-195: How would a footprint and track be affected if someone were walking forward, then looked behind them (in both directions) while continuing forward?

This is a good question and one that is beyond the scope of our current study.

• Another issue that relates to the hypothesis #3 interpretation is the difference between the shape of a single footprint and properties of a track (something measured among prints). These are different concepts to me, but I had the impression that "footprints" is used to cover both in this manuscript. This is relevant because the authors argue that Laetoli A differs from Laetoli G and S when track properties are considered (Fig 2D, Supp Figs 3D, 3A and 3B) but it is much less clear that the difference is present when the shape of a single footprint is considered (Supp Fig 3C, Supp Table 1 and actually even Supp Fig 4 gives that impression). For these reasons, I thought that the data presented came a bit short of demonstrating the authors' preferred interpretation—that this is the footprint of a grasping toe hominin.

We agree and have added footprint A2 to our comparison of footprint shape. With two prints considered, the results are even more robust. The Site A prints are as different from the Site G or S prints as a chimpanzee's footprint shape is from an average human foot.

o As a related point, the authors talk about the Burtele foot in relationship to footprint A3. It would be good to add something about how ray 1 length of the fossil and the footprint compare.

We mention the Burtele foot only in closing and not at all to suggest that the Site A footprints were made by this same species of hominin. In fact, the BRT-VP-2/73 foot has as exceptionally long Mt4 and a short hallux. While there is variation in relative toe length in extant apes and humans, the A3 footprint is not what we would necessarily expect given the known morphology of the Burtele fossil foot. This, in our opinion, remains an open question.

o It would be valuable for the authors to add a clear statement about whether they consider Track A to represent the "normal" gait of this other grasping toe hominin species or whether they consider this gait to be atypical (setting foot anatomy aside).

DARTMOUTH

This is a great question and one that is not possible to answer with the data we currently have. We have plans to return to Laetoli when it is safe to do so and prospect for additional footprint trails to assess, in part, how typical the Site A footprint trail is.

Smaller points:

Figure 1 – it would be nice to have a sense of the spatial relationship between Track A and Tracks G/S. Is it possible to include that in insets C or D?

This is a great idea and we have now adjusted the maps in Figure 1 to include the relative position of the sites.

Figure 1 – it would be useful to label the five footprints in the middle panels

Footprint labels have been added.

Line 118: check the figure call out (supposed to be Supp Fig 1 or 2?)

Line 124: check the figure call out (supposed to be Supp Fig 2?)

These figure call outs have been corrected.

Figure 2 Legend line 303: where bears walked four or more steps...; line 312 remove “with” possibly rephrase as ‘Ratio of stride width to step length across different...’

Suggested text changes made.

Figure 2B are these both left footprints?

No. The ursid print is from a right foot—the 5th digit is the longest, rather than the hallux. Text has been added to the figure legend to make this clearer: “If the Site A prints were made by an ursid, then print A3 would be a right print and the “hallux” impression would be from the 5th digit. If the Site A prints were made by a primate, then the A3 print would be from the left foot.”

Supplemental Table 1 – is it possible to include measures of variation?

Yes—standard deviations have been added where possible.

Supplemental Figure 3 – Are these comparisons all based on just one observation from the A3 print? What about the 5th print, which belongs to the same track but looks quite narrow?

Yes, these data are from the best-preserved footprint: A3. Note, however, that stride width and step length require multiple prints, and, in this case, the next best-preserved print (A2) is used. As detailed in the supplementary text, A5 is very poorly preserved and is from a part of the trackway that is severely eroded.

DARTMOUTH

In all figure and table legends, it would be useful to indicate whether the data come from chimps walking quadrupedally vs bipedally, or to indicate if these data are mixed. This info isn't readily apparent from the supplemental text.

Thank you for this recommendation. We have now added this information to the figure legends as recommended.

It would be good to include a sentence about how actual footprint measurements compare to those taken from pressure mat data.

We have added a sentence in the methods section recognizing that pressure mat data and footprint measurements do not always align (Hatala et al., 2013).

Lines 226-230: it would be useful to add a comparison of a chimp/human/Laetoli footprint to figure 3 (or Supp Figure 4) and one that ideally showing how a mid-tarsal break looks in footprint form.

In our experience, it is difficult to confidently and repeatedly identify a midtarsal break in a footprint made in a deformable substrate like wet volcanic ash or mud. We have therefore toned down our original assessment of the A3 foot as possessing a possible midtarsal break and instead suggest that there *may* be evidence for some midfoot mobility. Given this change, we don't think that a footprint with a midtarsal break (from a chimpanzee) would add to the paper.

Referee #3 (Remarks to the Author):

The manuscript titled "Footprint evidence for early hominin locomotor diversity – Laetoli, Tanzania" re-examines the Site A tracks which were originally discovered in 1976. The authors relocated and re-excavated the footprints in 2019, and conducted dimension-based assessments to determine if the tracks were hominin or if they could have been made by an ursid walking bipedally. In conclusion, they argue that the prints were likely made by a hominin. I commend the authors for their diligence and efforts in the relocation of the footprints first discovered and then reburied almost 45 years ago. I also applaud their data collection because it is quite an admirable sample of extant data which has been included in the study, alongside some very beautiful diagrams/figures.

Originality and significance: the manuscript is not 'traditionally' original because the prints were discussed in the 1970s, but the authors' relocation of the fossils, excavation and new analyses are of significance to the field. In palaeoanthropology, there are such few footprint data out there. If these prints are truly hominin then it would (1) boost our current dataset for hominin trackways, and (2) most importantly: could argue the case for two bipedal hominins during the Pliocene in Eastern Africa. The latter of which is very exciting.

We agree! Thank you to the reviewer.

DARTMOUTH

I have both minor and major comments of the study which I believe should be addressed prior to publication of the manuscript. The minor comments regard clarification of a few sentences, typographical errors, additional references to cite, and some necessary tweaks to a few figures for better readability. My major comments mostly regard the following: (1) the assumption that no erosion had occurred between 1976 and 2019, alongside a recommendation for an additional [quick] extra analysis to somewhat explore erosion of the bed/prints, (2) upon my examination of the Site A prints in the depth maps, I am unconvinced of the cross-cutting story and instead believe that these two prints may not belong to the same trackway – I recommend the authors conduct an MNI of the full trackway to determine the likelihood that all of these prints belong to the same individual and were made at the same time, (3) the ursid measurements must be included in the comparative analyses + statistics if the authors wish to establish if the prints are ursid or not, (4) I appreciate that the footprints are not in the best preservation condition, but as many footprints as possible must be included in the comparative analyses, there cannot be a sample size of one in the comparative analyses, and finally (5) the distance between the first and second digits is quite large and, therefore, I would urge the authors to calculate the hallucal angle for each of the Site A prints to determine if the angle falls within human/hominin ranges. This additional analysis would provide extra support to their conclusions.

These 5 major suggestions are repeated below, and we address them in detail there.

Minor comments:

Lines 143-146: I understand what the authors are trying to say, but I think that the wording of this sentence could be made clearer.

We did not make any changes to this text because it was not flagged by the other reviewers and efforts to improve the clarity made the sentence, in our opinion, more opaque. We are happy to take editorial suggestions to improve clarity.

On line 146 – please clarify the statement “no fossil ursids...”. It seems like a pre-emptive conclusion that the prints are not ursid, but hominin. Please clarify that you (assumedly) mean no skeletal material.

Text revised as suggested.

Line 147: “25,000 fossils and 85 mammalian species” – do the authors mean that there are 25000 fossils attributed to 85 mammalian species? Please clarify this statement.

We have modified the text as suggested.

Lines 156-157: please clarify if you mean the Site A or Site G prints.

Site A. Text correction made.

Line 158: “semi-wild juvenile black bears”, please specify species name. Please also provide ball-park

DARTMOUTH

figures for the statement that their foot sizes were similar in length to the track sizes. I appreciate that this can be found in the supp info, but it should also be included here using minimum amount of words.

We have added the species name and included average footprint lengths.

Lines 172-173 – Do the authors mean as in humans when taking short steps, or ursids? Please clarify.

Humans. We have modified the sentence to provide clarity.

Throughout manuscript: earlier in the manuscript, the authors state via footnote that they will use the terminology ‘hominin’ but interchangeably use the term ‘hominid’ and ‘hominin’. Because Nature has a wide readership (many of whom will presumably not be palaeoanthropology specialists), I urge the authors to either (1) change terminology to be consistent or (2) define why you are using the term ‘hominid’ versus ‘hominin’. The current footnote is not explanatory enough to a lay-reader.

We thank the reviewer and have modified several places where “hominin” is a better word-choice than “hominid” in this paper. While we use hominin to refer to humans and their many extinct relatives more closely related to *Homo* than *Pan*, the word “hominid” is still useful for us to include both humans and chimpanzees and we have added text to the footnote to make that clearer. When we write that the A3 footprint shape is more “hominid”-like than ursid-like, this communicates to the reader that the print is both human *and* chimpanzee-like, not solely “hominin”-like.

Lines 183: I’m personally not too sure what the phrase “scalloped impression” means as I have a few ideas of what this could look like when imagining it. Can a different terminology/phrase be used?

While we like the imagery of a scalloped impression, we have changed the text to read “semi-circular.”

Line 195: “we did not observe cross-stepping in our bear sample” – does this refer to quadrupedal steps or bipedal steps or both?

We only report data from bipedally walking bears. In this sentence, we have added “bipedal” before “bear”.

Paragraph beginning line 196: Are the authors referring to the pathological gait deformity called valgus knee? Or is it rather for the description of limb posture in hominins? If the former, then please provide reference(s) for valgus knee pathologies. If the latter, then please clarify in text that it is not a pathological description/interpretation. Perhaps a short explanation describing that the term valgus knee is related to the bicondylar angle/posture of the lower limb.

Valgus knee is a common term in the paleoanthropological literature and is not describing a pathological condition. We have simplified the language and introduced the term as, “a bicondylar angle (i.e., valgus knees).”

DARTMOUTH

Line 225: Can the figure/photograph of the footprint A3 in the supp info be cited at this point?

We have added, “Figure 3A; Supplementary Figure 2” to the text here.

Line 235: Please add “Dikika’, Afar, Ethiopia”.

Text added

Line 251: Please add the word ‘and’ between “distinct, presumably...”.

“and” added as suggested

Should the text after line 256 come under a new sub-heading?

We don’t think so since it is a short, concluding paragraph but we are happy to add a sub-heading if the editor requests it.

Line 291: Should Google Maps either have a citation or a (C) (copyright) symbol?

We have added the copyright symbol after “google” as suggested

Figure 2: the % times are reported in text, there is no need for panel A.

We disagree and find that a visual representation of data (as done here) complements and reinforces what is written in the text.

Please fix x-labels in panel D so that they are all aligned equally, i.e., “modern shod humans”. If it is difficult to align the text due to lengths of the labels, then consider shortening label names and clarifying in the figure caption. [this also applies to SI Figure 3 boxplots].

Thank you for this suggestion. We have modified the text so that it fits on a single line in both Figure 2D and Supplementary Figure 3.

Additionally, please re-iterate in the figure/figure caption the sample size for each group.

Thank you—we have added sample sizes to Figures 2C, 2D, to the legend of Figure 3B, and Supplementary Figures 3 & 4.

Panel B – the right ursid print seems to be cropped too closely to the border of the print and, as such, some data could be missing. Please see Falkingham’s 2014 paper on objective track outlines.

No cropping has occurred. This is an image of a 3d scan of a plaster cast of a black bear footprint.

DARTMOUTH

Panel C – please make this colour-blind friendly. Whilst the orange and the blue are fine, the blue on the black will not be readable by some (i.e., the median line in the boxplot).

Thank you for this suggestion. We have lightened the blue color so that the black line is clearer.

Line 371: please reword this sentence to state that it is not the same chimpanzee, but the same species (I assume?).

We do mean the same individual chimpanzees and have added “two” between “same” and “subadult” to make this clearer.

Line 371: MaxTraq and ImageJ need to be cited.

Thank you—we have added MaxTraq and Image J to our references.

Line 414: add space between “581individuals”

Space added.

Line 449 + 477: please specify if specific R packages were used? I assume Figure 3 was ggplot?

We have added the R packages to the text.

For clarity in the supp info, a table should be created listing the samples/specimens used. This would benefit the reader for quicker recap when examining the graphs.

Supplementary Table 1 includes this information.

Line 452: Because the analyses are a pivotal part of the paper, they should be briefly re-discussed here rather than referring the reader to another paper for the methods.

Thank you for this suggestion. We have now added a substantial amount of text to the methods section detailing how we were able to compare the Site A footprints to those made by humans, chimpanzees, and other hominins at Laetoli.

Line 473: for all Laetoli tracks? Or just the Site G or Site A? Please specify.

Yes, all. We have altered the text so that it reads: “For all Laetoli tracks (G, S, and A)...”

Line 485: please specify why the photographs were taken in the morning? Less glare from the sun? More shade?

DARTMOUTH

We have removed “in the morning” from this sentence but later in this paragraph we explain, “The images were all taken relatively early in the day, so there are significant shadows within each footprint that create strong contrasts.” Additionally, morning time avoids overheating of the equipment and the scientists!

In SI Figure 2: I like the diagrams! However, the blue shapes for digits 3, 4 and 5 haven't fully “closed”. Is it possible to remove the rogue blue lines extending from the shapes? Additionally, I think the arrows highlighting the midtarsal break put a bit too much emphasis on the ridge, whilst the manuscript contrastingly alludes to the ridge very briefly. The arrows should be replaced with a shape, a line or just one arrow to ‘soften’ the figure’s argument.

Thank you for this suggestion. We have redrawn Supplementary Figure 2 to fix these problems. Instead of an arrow, we have softened the argument by adding a transparent red oval. Additionally, here and throughout we refer to this as midfoot mobility rather than a midtarsal break since midfoot mobility is a continuous feature of the hominid foot.

I very much like the supplementary pdf of the 3D model. However, can the underside of the model be given more colour? Currently when I rotate the model using Adobe, I am unable to clearly see the underside due to the lighting. Will the model also be made available upon publication to download?

We thank the reviewer for noticing this. This is a lighting/settings issue that can be resolved by telling Adobe Reader/Acrobat to render the underside of the model by opening the preferences dialogue, going to “3d+Multimedia” and checking “enable double sided rendering.” Our understanding is that this cannot be changed by us in the pdf to be done automatically.

The Miocene footprints from Crete also investigated if the prints were hominin or ursid. This paper should be referenced here (Gierlinski et al. 2017; Crompton 2017).

Thank you. We have added supplementary text and these references.

Major comments:

As I am understanding it, based upon qualitative assessment of the tracks, an assumption was made that little to no erosion occurred? Whilst I agree that the data does not exist to quantitatively assess erosion of the prints, I do not quite agree with the statement that no erosion occurred. This statement should be softened and acknowledged that there is scope for some erosional processes to have occurred as has been studied elsewhere (citing Zimmer et al. 2018; Wiseman et al. 2018 whom both quantitatively examined erosional processes in footprints). In the supplementary information the authors discuss how their measurements vary versus those collected in the 70s by Tuttle. All measurements now are smaller than Tuttle’s. This appears to me to be the cause of erosion, although a stronger rebuttal and additional evidence from the authors could be persuasive enough to challenge my concern – for example, a quick mesh to mesh distance analysis either in R or CloudCompare could be computed using the 3D models to visually examine the shape differences between 1976 and 2019 (with an explanation for the regions that have had additional debris removed).

DARTMOUTH

We thank the reviewer for this suggestion (though I believe the reviewer meant “larger” rather than “smaller” for our measurements compared with Tuttle). We have adopted the recommendation and assessed erosion using Cloud Compare. We have added these results to Supplementary Figure 1 and to the supplementary text. In short, we find that some erosion has occurred around the rim of an oval depression (pothole) positioned just west of the Site A trail. However, the same pattern of difference cannot be found around the rim of the A2 and A3 prints and instead, the major differences between the 1977 and 2019 photogrammetry meshes exists internally. The best explanation for these differences is that we conducted a more thorough excavation of the prints in 2019 compared with 1977.

Figure 1: Panels D and E are hard to read with the white text on the black background. The colours should be inverted for ease of readability. Also, it is not entirely clear what the orange rectangle in panel D is showing – I presume it is highlighting where panel E is located? Perhaps panel E could become a ‘pop-out’ box (i.e., in which lines connect the edges of the rectangle to panel E) from panel D for clarity? Additionally, the text in panel C needs to be bigger.

Thank you for these suggestions. We have redesigned panels A-C of Figure 1 to increase readability, make clear which panels are pop-outs, and show the relationship between Laetoli site A and the other bipedal prints at G and S.

Is the example of cross-stepping shown in the middle panels? If so, those cross-steps are quite extreme and rather look like they could belong to different individuals. How sure are the authors that this is cross-stepping? Do the footprints show evidence of slippage which could correspond to cross-stepping? Assuming that the base of support lies medially to the right footprint, that cross step is what I would expect the maximum bounds in an anatomically modern human; I’m sceptical that is a cross-step in a hominin which may not have adducted hips + valgus knee, particularly without additional trace evidence of slippage. This argument requires further evidence to back up the claim.

We have added additional data examining the effects of cross-stepping on footprint morphology in a sample of humans (N=10). We are puzzled that the reviewer assumes that the hominin would not have adducted hips and a valgus knee.

Otherwise, the authors should soften their argument and state that it could either be cross-cutting evidence or could potentially be footprints belonging to two individuals not within the same trackway.

See below

The right footprint also looks a bit problematic in which the heel looks a bit too pointed and toe line too straight – which leads me back to my original comment: how sure are the authors of minimal erosion of the prints?

DARTMOUTH

See above. We have used Cloud Compare to quantify erosion and found little evidence that this can explain the morphology of the footprints. However, we wonder if the reviewer may be looking at print A4 which is quite eroded and therefore does not factor much into our analysis and our interpretations.

But also – are the authors certain that the shapes of the footprints are hominin and not ursid? Can additional analyses be computed to convince the reader, such as shape analyses?

We feel as though we have sufficiently refuted the ursid hypothesis (see Figure 2; Supplementary Figure 3; Supplementary text “*Agriotherium*”). Our shape analysis (Figure 3) finds the footprints to be similar in shape to chimpanzees.

I would also argue for the presence of an unidentified footprint in that middle panel – there appears to be a print crossing almost diagonally across the top. The shape of the print on the colour-rendered map is very similar to human prints from Namibia and Formby Point. Has the diagonal impression been investigated?

Yes. It is a small (10 cm long) pothole with eroded edges. It is perhaps related to the fault line crack extending northwest from the tip. None of us in the field identified this as a footprint, and certainly not a hominin one.

If so, then I trust the opinion of the authors whom have the 3D data in front of them, but (if so) I would also urge the authors to add an additional section to the supp info on other hollows in the area. For example, could the other impressions on the right side of the panels also be potential prints?

On the right are a number of lagomorph and small artiodactyl footprints. These were identified by Leakey and Harris and confirmed by us. Occasionally, two consecutive small mammal prints yield the oblong shape that can superficially resemble a hominin footprint but are not. Additionally, along the right panel is the edge of a 45 cm wide proboscidean print. We examined every inch of this area in detail and to the best of our knowledge/expertise located all the hominin footprints present.

And what methods did you use to exclude them from the conclusion that a single trackway was present? My opinion based upon these images is that there are more than one individual present and this needs to be investigated.

We welcome additional interpretations of the Site A footprints. However, based on the many days we spent with these footprints in the field, we found no evidence that there is a second individual present. Footprints A1-A3 are the same length (160-165 mm); A4 and A5 are too badly eroded to measure length but share similar overall size to A1-A3. If, as the reviewer proposes, A2 and A3 are different individuals, where is the left foot of the A2 individual? And where is the right print from the A3 individual? It seems to us a *much* simpler explanation that this is a single individual cross-stepping.

What is the MNI based upon measurements/shape?

DARTMOUTH

One. All of the reliable measurements yield similar values. The shape analysis (Figure 3) yields a shape of A2 and A3 that are quite similar to one another.

Finally, I would recommended to explicitly highlight the tracks on the middle panels for clarity. Towards the top of the panel they become a bit harder to discern and I think the reader would benefit from them being highlighted/circled/numbered/etc.

Thank you. We have added labels identifying the prints to the middle panels of Figure 1.

Figure 3: because this paper is investigating if the prints were ursid or not, ursid measurements need to be included in this figure + analysis. In panel A – has the site G photograph been reproduced from another publication? Or used with permission from someone? Please state in figure caption if so. Additionally, please rearrange the labels in panel C. It is currently hard to read “Laetoli G” for example, due to the label’s close proximity to the dashed lines.

We disagree. Figure 2 is meant to assess the ursid hypothesis. After that was sufficiently refuted, the question became what hominin the trackway is from. Figure 3 was deliberately designed with that question in mind and we fear that including ursids in that figure will unnecessarily crowd the figure. The Site G photograph was taken by J. Reader (a co-author of the paper). However, it is property of Science Photo Library and we have now credited them in the figure legend. Figure 3c labels have been modified as requested.

Apologies, but I have to disagree with the authors that that black bear and Site A prints are similar in size. 145mm versus 161mm is quite a difference; in human children this is a difference of 5 shoe sizes (i.e., assuming half sizes are included – but could be argued for just two shoe sizes difference –so a size 7 versus a size 9; measurements calculated using startriteshoes.com/size-calculator). I do not argue that this discrepancy effects their analyses, but the wording should be changed. Additionally, because the analyses rely on the fact that feet are so similar in size, I would recommend that the data should be normalised prior to statistical analyses (sufficient to use ratios).

When we collected these data on bear footprints, we were using foot lengths reported by Tuttle (~141 mm) and our bears were only a tad longer (~145mm). However, when we fully excavated the Site A footprints, we found that the trackway consisted of slightly longer prints than originally reported (~161mm). However, this difference is still within 10% of the size of the bear prints and thus we find them to be comparable, albeit not identical. We have altered the wording of the text and report that the prints are “within 10% of the length” of one another.

Line 456: A sample size of one is insufficient to be conducting analyses, and I am surprised by the authors’ decision to exclude most of a readily available sample. Despite the other tracks being of ‘poorer quality’, as much data as possible from Site A must be included. From visual inspection of the figures, I would argue that at least three of the prints are usable. Otherwise, the morphology from this one print could be driven by erosion, poor definition, or just simply an ‘anomaly’ of a print. McClymont and

DARTMOUTH

Bennett (2021) (see also McClymont's PhD thesis; U of Liverpool 2016) established that a sample size of >250 is necessary to make inferences from footprints. But please also see Belvedere et al. (in press; Palaeontology – Questions of sample in ichnology; please contact authors if not yet published by the time revisions are being done for paper sharing) in which a minimum sample size of seven was required to make inferences from footprints based upon change-point modelling. Whilst I completely understand that this is an impossible sample size for the current study due to only five fossil footprints being present, I must urge the authors to include more than one footprint in their comparative analysis here. This is particularly pertinent due to the analysis regarding depth measurements – some footprints are light, some footprints are deep. With a sample size of just one, then the spectrum is lost. More-so, if there is evidence of cross-stepping then I would assume that the plantar pressure on the cross-stepping foot would be greater, thus skewing results. If there is any slippage in the footprint then this may further affect the results. To give a broad overview into the Site A footprints (for the first time) then more footprints must be included in the comparative analyses.

We thank the reviewer for this suggestion and in response we included print A2, in addition to A3, to our analysis. The prints are remarkably similar to one another resulting in a more robust argument for the Site A footprints being as different from the Site G or S prints as chimpanzee foot shapes are from a human foot. This finding also challenges the reviewer's suggestion that the MNI is >1. We finally note that the other footprints (A1, A4 & A5) are not preserved well enough to include them in this shape analysis.

Regarding the depths of the prints, please state how the depths were measured. Was this the absolute depth from the surface? How was this defined? Was a plane created?

Yes. We have now added to the methods section the following,

“For each experimental and fossil footprint, 3-dimensional models were constructed using photogrammetry, through a variety of methods described here for Laetoli Site A and elsewhere by the authors for other samples^{20,21,35,41}. Using Geomagic Wrap 2021 (3D Systems, Rock Hill, SC, USA), a best-fit plane was fit to the undisturbed substrate surrounding each track, and this was fixed to the XY plane in world coordinate space. In this orientation, depths of the footprint were measured in the regions of the medial and lateral heel, medial and lateral midfoot, and all five metatarsal heads and toes. Raw depth measurements were normalized, within each footprint, to a scale of 0 to 1 in order to compare the topologies of footprints that may vary in depth.”

Line 466: in the creation of the mean human footprint, please cite Bennett et al. 2016 and Belvedere et al. 2018 in which both of those papers created an averaged track. Please also clarify what is meant by stating that it is not an actual footprint? Is it composed of different measurements? Shape differences could be quantified using DigTrace Pro/Academic which is freely available software and could provide an averaged 3D track as a .obj file. That 3D model could then be compared to the Site A print via the same method of track registration.

DARTMOUTH

Our methods for creating an average track were not clear and we have therefore revised this section to now read,

“An overall “human mean footprint” was also computed by averaging the within-subject mean normalized depths, and this represented a measure of central tendency as described below.

To represent the range of observed variation in human footprint topography, for 10,000 iterations we randomly sampled a human subject and drew a sample of two of their footprints. We then averaged the normalized depths of those two footprints and computed the Mahalanobis distance (using the between-subject covariance matrix) between this track and the mean of all other subjects’ footprints. Also, for 10,000 iterations we selected a random chimpanzee subject, drew a random sample of two of their footprints and computed the Mahalanobis distance between the average of those tracks and the overall mean human footprint. For the Laetoli tracks, Site A and Site S samples only included two tracks, so these were simply averaged and the Mahalanobis distance was calculated between each averaged track and the mean human footprint. For Laetoli Site G, all possible two-track combinations were drawn from the sample described above, and the Mahalanobis distance was calculated between the averaged track from each combination and the human mean. In all cases, we calculated multivariate distances using the human between-subject covariance matrix (i.e., treating the chimpanzee and fossil tracks as if they came from different human subjects). All analyses described above, and the histogram displaying multivariate distances (Fig. 3), were generated using R (v. 3.6.1).”

Regarding the photogrammetry: Please state what different software was attempted to reconstruct the prints – I assume more than one? Although I do applaud the authors for managing to create a 3D model in Agisoft. Peter Falkingham has a great online (free) blog of different photogrammetry techniques and software that can be used (<https://peterfalkingham.com>). Some software computes un-distortion of images which may produce more precise models. Additionally, please compare the older 3D models with the re-excavated versions (perhaps even using a simple mesh-to-mesh distance analysis) to examine how similar/dissimilar the prints are. This may assist in identifying erosion.

As reported in the “Photogrammetry” section, we used Agisoft Photoscan Pro-Metashape Pro and we are pleased with the results. As recommended above, we have now added the results of an erosion analysis using Cloud Compare and find that while some rim erosion occurred to a depression at the site, the statistically significant differences between the 1977 and 2019 Site A tracks are almost entirely within the prints themselves, meaning that differences are the result of proper excavation of the prints rather than print erosion.

Line 602: 13mm between the first and second digits is quite large! The paper does not discuss hallucal ad/abduction, but I urge the authors to incorporate this into their manuscript. I would also be interested to know the ranges of hallucal angles in the Site A prints and how they compare with the other Laetoli

DARTMOUTH

prints but also with humans. I believe that is a stronger story than inferences upon knee valgus which cannot be supported without biomechanical assessments.

Thank you for this suggestion. We attempted to calculate the abduction angle as measured by Bennett et al. but could not since matrix obscures the deepest region of the ball of the foot under the first metatarsal head and substantially increased error. That is the reason that we developed this original means of measuring the distance between the first and second digits. This information has now been added to the legend of Supplementary Figure 4.

SI Figure 3: in panel D please either change the size, colour or shape of the icons representing the Laetoli samples. It was quite difficult to easily locate Site A on the graph (and in fact I only found it after reading the figure caption), and the other shapes were equally as hard to see too. I would also be careful about making too many strong assumptions based upon this graph. There is overlap amongst all groups on that graph, and the sample sizes for the Laetoli prints is quite small. Additionally, the negative value for the cross-stepping is problematic. I'm not convinced of the cross-stepping story and would like to see more evidence to support it. If cross-stepping can be supported with more evidence, then I wouldn't see the issue in using absolute values in this graph so long as it was clearly stated in the figure caption that you have done so. Otherwise, this graph does not contribute much to the story of the paper in which the reader wants to see where Site A falls with respect to the other taxa.

We thank the reviewer for this suggestion and have adjusted the colors of the markers in Supplemental Figure 3 to try increase readability. While we appreciate that this graph is not terribly revealing, we do find that it communicates why the Site A footprints have been so difficult to interpret and we would like to keep it in the supplemental part of the paper.

SI Figure 4 and the accompanying analyses should include ursid data.

We disagree. Ursids do not possess a grasping hallux nor did they descend from an ancestor with one. We do not think that adding ursids to this plot would be informative. Further, if the Site A footprint trail was from an ursid, then print A3 would be from a right foot, not a left, making the "hallux" a fifth digit. Thus, we would be comparing the divergence of the first digit in primates to the fifth in ursids which would be an apples-oranges comparison.

Overall, I enjoyed reading the paper and especially enjoyed the detailed story about the re-excavation of the prints which is often lacking in other papers. I do believe that this paper is publishable after revisions.

Thank you for your insightful comments and helpful suggestions!

Reviewer Reports on the First Revision:

Referee #1 (Remarks to the Author):

This manuscript is improved in many ways. But I still have three lingering suggestions/concerns. First, the major argument of this paper is that two different hominin species are represented at Laetoli but a very large percentage of the paper tests the weak hypothesis that the footprints are ursid. This was never a compelling hypothesis and I would advocate that this entire section be moved to supplementary materials. In my opinion, it's a waste of precious space and time to read this "straw bear" argument in detail.

Second, I was glad to read in the rebuttal that A2 was added to A3 in the multivariate analysis, but I found little evidence of that addition in the paper including the analysis presented in Figure 3. Am I missing something? Also, while the addition of cross-stepping humans is a welcome addition, the big question is whether footprints in a cross-stepping afarensis-like hominin who made the G tracks might be different from the A track. That issue remains untested. Also, if it was cross stepping because of some perturbation, shouldn't the experiment look at cross-stepping in humans experiencing some kind of perturbation?

Finally, leaving aside that the basic argument of this paper still comes primarily from one or possibly two very chimplike footprints, I would urge the authors to discuss the significance of the very chimplike nature of the footprints. The possibility that something like Burtele made them now appears to be expunged from the ms (which is unwise), and the ms lacks any discussion of the biological implications of a bipedal hominin whose footprints cannot be distinguished from a bipedal chimpanzee (except in terms of step length to width leaving aside cross-stepping) also being present at Laetoli. Essentially, the paper lacks a discussion section. What do the authors think?

Referee #2 (Remarks to the Author):

In the revised manuscript, the authors added an experiment to address the question of how cross-stepping would affect footprint shape (Supplementary Methods- Human cross-stepping footprint experiments). That experiment addresses my initial comment precisely and goes a long way to strengthen the argument that Track A was made by a different species than Track G or S. The only small suggestion I would have is in regard to how the data from the new experiment are incorporated into the manuscript. Supplementary Figure 5 shows the overall magnitude of the shape difference, but doesn't allow the reader to assess the nature of the difference. Brief statements are made in the SOM text, but it would be informative to have the human cross-stepping data reported in Supplemental Table 1 and shown in Supplementary Figures 3-4. Other than that, the revisions thoroughly addressed my comments and the manuscript is ready for publication.

Referee #3 (Remarks to the Author):

The authors present their revised manuscript discussing possible hominin trackways at Laetoli. I am happy that the authors have addressed most of my concerns, and I am particularly pleased to see that a further footprint from Site A has now been incorporated into their analyses and that their results are upheld - very promising and exciting! I am additionally pleased to see additional experiments have been included to address the cross-stepping conundrum. Finally, I would like to thank the reviewers for investigating possible erosion in the prints, and for addressing my

comments. I have some further very minor comments, after which I believe that the manuscript will be ready for publication:

Thank you for changing Figure S2, however, the shapes are still not closed (if you zoom in on digits 2-5, you can see blue off shoot lines where the holes haven't closed fully). Whilst this isn't necessarily a 'revision' for the authors, I would highly recommend making this edit because all other figures are perfectly designed, it would be a shame for this one to slip through.

Great job on recreating Figure 1 - looks good!

I think my previous comment may not have been clear regarding valgus knees and adducted hips. I was not implying that the hominin was unlikely to have had such anatomies, but rather that the language used to describe such anatomies was quite strong, despite little evidence to back such a claim (and I'm not advocating for them to do so). Rather that the language could be softened until further biomechanical analyses are conducted to investigate these claims that the hominid did not have abducted hips etc (which are well beyond the scope of the current paper). However - I commend the authors on the additional analyses which they have conducted, and that these edits do sufficiently tackle my earlier concerns.

I'm still not fully sold by the cross-stepping story, but I believe that the authors have argued their case well, and that no further corrections regarding cross-stepping are required. This can be explored in future studies.

Minor comments: line 229 needs a reference to back up the claim, perhaps Shu et al. 2015?

Line 262: consider removing *A. afarensis*, and replacing with either the genus name or just 'hominin'. The authors very eloquently spend the previous paragraphs debating who could have made the prints, and stating that it is not a strong case for *afarensis*, yet this one sentence here seems out of place - almost as if the authors have changed their mind on taxonomic association, and then change it again in the following sentences.

Recommendation: The color map 'viridis' is colorblind friendly and the recommended choice for the mesh to mesh distance analysis shown in Figure S1 (it is a pre-loaded colormap available with CloudCompare). Parula (the current colormap) is not colorblind friendly, unfortunately.

Overall, I recommend publication of this manuscript following minor revisions and do not believe that this manuscript will be needed to be returned to review after such minor revisions. Good job to the authors!

Author Rebuttals to First Revision:

Referee #1 (Remarks to the Author):

This manuscript is improved in many ways. But I still have three lingering suggestions/concerns. First, the major argument of this paper is that two different hominin species are represented at Laetoli but a very large percentage of the paper tests the weak hypothesis that the footprints are ursid. This was never a compelling hypothesis and I would advocate that this entire section be moved to supplementary materials. In my opinion, it's a waste of precious space and time to read this "straw bear" argument in detail.

We appreciate this perspective and agree that in retrospect, the “straw bear” argument is not compelling. This hypothesis was proposed with the best information available at the time and it stood uncontested for over 30 years. We have shortened this part of the paper, but believe that it is a crucial part of the narrative of the Site A footprints and deserves the text we devote to it in the main part of the paper.

Second, I was glad to read in the rebuttal that A2 was added to A3 in the multivariate analysis, but I found little evidence of that addition in the paper including the analysis presented in Figure 3. Am I missing something? Also, while the addition of cross-stepping humans is a welcome addition, the big question is whether footprints in a cross-stepping afarensis-like hominin who made the G tracks might be different from the A track. That issue remains untested. Also, if it was cross stepping because of some perturbation, shouldn't the experiment look at cross-stepping in humans experiencing some kind of perturbation?

Yes, A2 is now included in Figure 3c—the value reported is a mean of two prints (A2 and A3). This is stated in the legend of Figure 3. Humans are the only living hominin available to test whether or not cross-stepping produces significant variation in footprint metrics. We conducted this experiment to test whether the differences in footsteps found between humans walking with and without cross-stepping are similar in magnitude and direction to the differences in the footprints left at sites G and A. They are not, which is one of many pieces of evidence that the Site A trackway was not made by *A. afarensis*. Perturbation is one potential reason why humans cross step, however we are not arguing that this explains the Site A prints. We thank the reviewer for this suggestion, which is worthy of future study.

Finally, leaving aside that the basic argument of this paper still comes primarily from one or possibly two very chimplike footprints, I would urge the authors to discuss the significance of the very chimplike nature of the footprints. The possibility that something like Burtele made them now appears to be expunged from the ms (which is unwise), and the ms lacks any discussion of the biological implications of a bipedal hominin whose footprints cannot be distinguished from a bipedal chimpanzee (except in terms of step length to width leaving aside cross-stepping) also being present at Laetoli. Essentially, the paper lacks a discussion section. What do the authors think?

While the footprints are chimpanzee like in some respects (Figure 3c), they are hominin-like in others (e.g. bipedal gait; less divergent hallux; evidence for extended knee and hip kinematics from toe depth ratio). We agree, however, that the more primitive (e.g. chimpanzee-like) aspects of the foot and its superficial similarities to the Burtele foot are fascinating and worthy of continued study. However, we caution that without more fossil or footprint evidence than what we currently have, any effort to directly connect the Site A footprints to the Burtele foot is too speculative. Indeed, we are not arguing that the Burtele hominin is involved, only that the discovery of the Burtele foot is one of many expanding lines of evidence for increased locomotor diversity in the Plio-Pleistocene.

Referee #2 (Remarks to the Author):

In the revised manuscript, the authors added an experiment to address the question of how cross-stepping would affect footprint shape (Supplementary Methods- Human cross-stepping footprint experiments). That experiment addresses my initial comment precisely and goes a long way to strengthen the argument that Track A was made by a different species than Track G or S. The only small suggestion I would have is in regard to how the data from the new experiment are incorporated into the manuscript. Supplementary Figure 5 shows the overall magnitude of the shape difference, but doesn't allow the reader to assess the nature of the difference. Brief statements are made in the SOM text, but it would be informative to have the human cross-stepping data reported in Supplemental Table 1 and shown in Supplementary Figures 3-4. Other than that, the revisions thoroughly addressed my comments and the manuscript is ready for publication.

Thank you for the suggestion. We have incorporated the cross-stepping data into Supplementary Table 1. We have not included them in Extended Data Fig. 4 [formerly Supp. Fig. 3] due to space constraints nor Extended Data Fig. 5 [formerly Supp. Fig. 4], as we did not collect divergence ratio data for our cross-stepping sample and it is unlikely that cross-stepping would significantly change the divergence of the hallux.

Referee #3 (Remarks to the Author):

The authors present their revised manuscript discussing possible hominin trackways at Laetoli. I am happy that the authors have addressed most of my concerns, and I am particularly pleased to see that a further footprint from Site A has now been incorporated into their analyses and that their results are upheld - very promising and exciting! I am additionally pleased to see additional experiments have been included to address the cross-stepping conundrum. Finally, I would like to thank the reviewers for investigating possible erosion in the prints, and for addressing my comments. I have some further very minor comments, after which I believe that the manuscript will be ready for publication:

Thank you for changing Figure S2, however, the shapes are still not closed (if you zoom in on digits 2-5, you can see blue off shoot lines where the holes haven't closed fully). Whilst this isn't necessarily a 'revision' for the authors, I would highly recommend making this edit because all other figures are perfectly designed, it would be a shame for this one to slip through.

Thank you for the suggestion. We have edited the shapes highlighting the digits.

Great job on recreating Figure 1 - looks good!

I think my previous comment may not have been clear regarding valgus knees and adducted hips. I

was not implying that the hominin was unlikely to have had such anatomies, but rather that the language used to describe such anatomies was quite strong, despite little evidence to back such a claim (and I'm not advocating for them to do so). Rather that the language could be softened until further biomechanical analyses are conducted to investigate these claims that the hominid did not have abducted hips etc (which are well beyond the scope of the current paper). However - I commend the authors on the additional analyses which they have conducted, and that these edits do sufficiently tackle my earlier concerns.

I'm still not fully sold by the cross-stepping story, but I believe that the authors have argued their case well, and that no further corrections regarding cross-stepping are required. This can be explored in future studies.

Minor comments: line 229 needs a reference to back up the claim, perhaps Shu et al. 2015?

Thank you for the suggestion. Our statement is in reference to the data presented in Figure 3. We agree that these data are consistent with Shu et al., 2015 and additionally D'Aout et al., 2009 & Musiba et al., 1997.

Line 262: consider removing *A. afarensis*, and replacing with either the genus name or just 'hominin'. The authors very eloquently spend the previous paragraphs debating who could have made the prints, and stating that it is not a strong case for *afarensis*, yet this one sentence here seems out of place - almost as if the authors have changed their mind on taxonomic association, and then change it again in the following sentences.

Thank you for the comment. We have made some edits to this section but have kept "*A. afarensis*." We have presented evidence that Site A was made by a cross-stepping hominin. This section presents the final evidence rejecting the possibility that the unusual characteristics discussed in the proceeding paragraphs could be explained as the result of changes in footprint morphology due to an *A. afarensis* cross-stepping. Here we present evidence derived from a human study that the difference between the Site G and A footprints is not similar in magnitude or direction to the difference between cross-stepping and non cross-stepping humans.

Recommendation: The color map 'viridis' is colorblind friendly and the recommended choice for the mesh to mesh distance analysis shown in Figure S1 (it is a pre-loaded colormap available with CloudCompare). Parula (the current colormap) is not colorblind friendly, unfortunately.

Thank you for your comment. We have redone that image to make it more readable to those who are color blind.

Overall, I recommend publication of this manuscript following minor revisions and do not believe that this manuscript will be needed to be returned to review after such minor revisions.
Good job to the authors!